# Room temperature Planar Hall effect in nanostructures of trigonal-PtBi$_2$

Arthur Veyrat,[1, 2, 3] Klaus Koepernik,[1, 2] Louis Veyrat,[1, 2, 4] Grigory Shipunov,[1, 2] Saicharan Aswartham,[1, 2] Jiang Qu,[1, 2] Ankit Kumar,[1, 2] Michele Ceccardi,[5, 6] Federico Caglieris,[6] Nicolás Pérez Rodríguez,[1, 2] Romain Giraud,[1, 2, 7] Bernd Büchner,[1, 2, 8] Jeroen van den Brink,[1, 2, 8] Carmine Ortix,[9] and Joseph Dufouleur[1, 2, 10, *]

[1] *Leibniz Institute for Solid State and Materials Research (IFW Dresden), Helmholtzstraße 20, D-01069 Dresden, Germany*
[2] *Würzburg-Dresden Cluster of Excellence ct.qmat, Dresden, Germany*
[3] *Laboratoire de Physique des Solides (LPS Orsay), 510 Rue André Rivière, 91400 Orsay, France*
[4] *CNRS, Laboratoire National des Champs Magnétiques Intenses, Université Grenoble-Alpes, Université Toulouse 3, INSA-Toulouse, EMFL, 31400 Toulouse, France*
[5] *Department of Physics, University of Genoa, 16146 Genoa, Italy*
[6] *CNR-SPIN Institute, 16152 Genoa, Italy*
[7] *Université Grenoble Alpes, CNRS, CEA, Grenoble-INP, Spintec, F-38000 Grenoble, France*
[8] *Department of Physics, TU Dresden, D-01062 Dresden, Germany*
[9] *Dipartimento di Fisica "E. R. Caianiello", Universitá di Salerno, IT-84084 Fisciano (SA), Italy*
[10] *Center for Transport and Devices, TU Dresden, D-01069 Dresden, Germany*

**Trigonal-PtBi$_2$ has recently garnered significant interest as it exhibits unique superconducting topological surface states due to electron pairing on Fermi arcs connecting bulk Weyl nodes. Furthermore, topological nodal lines have been predicted in trigonal-PtBi$_2$, and their signature was measured in magnetotransport as a dissipationless, *i.e.* odd under a magnetic field reversal, anomalous planar Hall effect. Understanding the topological superconducting surface state in trigonal-PtBi$_2$ requires unravelling the intrinsic geometric properties of the normal state electronic wavefunctions and further studies of their hallmarks in charge transport characteristics are needed. In this work, we reveal the presence of a strong dissipative, *i.e.* even under a magnetic field reversal, planar Hall effect in PtBi$_2$ at low magnetic fields and up to room temperature. This robust response can be attributed to the presence of Weyl nodes close to the Fermi energy. While this effect generally follows the theoretical prediction for a planar Hall effect in a Weyl semimetal, we show that it deviates from theoretical expectations at both low fields and high temperatures. We also discuss the origin of the PHE in our material, and the contributions of both the topological features in PtBi$_2$ and its possible trivial origin. Our results strengthen the topological nature of PtBi$_2$ and the strong influence of quantum geometric effects on the electronic transport properties of the low energy normal state .**

## INTRODUCTION

Topology is, together with superconductivity, one of the most striking macroscopic manifestation of the quantum nature of electrons in quantum materials. A consequence of the geometric properties of the wave function, from which topological properties arise, is the existence of the Berry curvature (BC): an emerging magnetic field in momentum space. In a Weyl semimetal, the Weyl nodes act as sources and sinks of BC [1] and can lead to signatures in charge-transport experiments such as the planar Hall effect (PHE) [2–4]: the appearance under an external in-plane magnetic field of a transverse voltage dependent on the relative orientations of the current and magnetic field. The PHE is dissipative, i.e. it is symmetric in magnetic field and associated with an in-plane field-dependent longitudinal voltage called anisotropic magnetoresistance (AMR) [5]. However, mechanisms different from the BC such as intrinsic or orbital magnetism can also result in a dissipative PHE and AMR, making it an ambiguous signature of quantum geometric properties [6, 7].

Here, we study the PHE in magnetic field and temperature to find new insight on the quantum geometric properties in trigonal-PtBi$_2$: a non-magnetic Weyl- and nodal-line-metal which was recently reported in transport experiments to exhibit sub-Kelvin 2D-superconductivity and a BKT transition in nanostructures [8]. Scanning tunnelling spectroscopy (STS) studies in this compound have reported higher-temperature surface superconductivity [9], which was confirmed by angle-resolved photoemission spectroscopy (ARPES) to be intrinsically – and solely – generated by topological Fermi arcs [10] (The presence of Fermi arcs was also confirmed by STS measurements [11]). This makes t-PtBi$_2$ a very promising candidate for intrinsic topological superconductivity.

A thorough investigation of the transport signatures of topological degeneracies in t-PtBi$_2$ are however still lacking. Here, we focus on the bulk properties of t-PtBi$_2$ nanostructures and report on a robust dissipative PHE from low magnetic fields and up to room temperature. These results are entirely coherent with the predicted Weyl nature of t-PtBi$_2$.

## PLANAR HALL EFFECT

### Planar Hall effect in Weyl Semi-metals

The planar Hall effect is a generic signature of the anisotropy of the magnetoresistivity that uniquely depends on the relative orientations of the in-plane magnetic field and electric field (i.e. current) in the sample. It thus corresponds to a difference $\Delta\sigma = \sigma_\parallel - \sigma_\perp$ between the conductivity $\sigma_\parallel$ when the two fields are aligned, and the conductivity $\sigma_\perp$ when they are orthogonal. As a result, it manifests itself both in transverse and longitudinal resistance measurements (the latter being sometimes referred to as anisotropic magnetoresistance, AMR, or anisotropic longitudinal magnetoconductance, LMC). Since the two effects (PHE and LMC/AMR) are in a one-to-one correspondence linked, we will refer to them both as PHE from here on. The planar Hall effect has historically been predicted and discovered in ferromagnets [12, 13], where the anisotropy is caused by the concomitant presence of intrinsic magnetization of the material and spin-orbit interaction. In materials without long-range magnetic order, a giant PHE has been predicted to occur in Dirac and Weyl semi-metals as a direct consequence of the chiral anomaly [2, 3]. This results in a large negative contribution to the longitudinal magnetoresistance when the magnetic and electric fields are aligned [2, 14]. In the simple case of a type-I Weyl cone (i.e. when the Weyl cone is not tilted), the contributions of the PHE to the longitudinal conductivity $\sigma_{xx}$ and the transverse conductivity $\sigma_{yx}$ follow the angular dependence [3]:

$$
\begin{aligned}
\sigma_{xx}^{PHE}(\varphi) &= \sigma_\perp + \Delta\sigma \, \cos^2\varphi, \\
\sigma_{yx}^{PHE}(\varphi) &= \Delta\sigma \sin\varphi \cos\varphi,
\end{aligned}
\tag{1}
$$

with $\Delta\sigma = \sigma_\parallel - \sigma_\perp$ ; $\sigma_\perp = \sigma_D$ the Drude conductivity, independent of the magnetic field ; $\sigma_\parallel = \sigma_D + a \times B^2$ with $a$ a constant which depends on the carrier group velocities and the BC, and $B$ the amplitude of the in-plane magnetic field ; and $\varphi$ the relative angle between the magnetic and electric fields in the sample. As can be seen from Equation 1, the PHE is characterized by a $\pi$-periodic oscillation of both the longitudinal and transverse conductivities, when the magnetic field is rotated in-plane (while keeping the current direction fixed), with a $\pi/4$ offset between them, and the amplitude of the oscillations is expected to increase quadratically with magnetic field.

When the Weyl cones (WC) acquire a tilt (or an overtilt in the case of type-II Weyl cones), the field dependence of the PHE depends on the orientation of the current: when the current flows along the tilt direction, the amplitude of the PHE is expected to grow linearly for small magnetic fields. When the current flows perpendic-

ular to the tilt direction however, the PHE is expected to grow quadratically with field, precisely as for non-tilted Weyl cones [3]. In real materials, the tilt vector can be pinned to principal crystallographic directions by multiple point-group symmetries (e.g. mirror symmetries). However, in materials with low-symmetry content the tilt vector will not be aligned with the crystallographic axes.

### Resistivity versus conductivity

We first recall that in magnetotransport experiments, we have no direct access to the conductivity. Instead, we measure the resistance of the sample, which is linked to the resistivity through the geometry of the sample. The behaviour of the resistivity contribution of the PHE is actually expected (in the type-I case at least) to follow very closely that of the conductivity [2, 6]:

$$
\begin{aligned}
\rho_{xx}^{PHE}(\varphi) &= \rho_\perp - \Delta\rho \, \cos^2\varphi, \\
\rho_{yx}^{PHE}(\varphi) &= -\Delta\rho \sin\varphi \cos\varphi,
\end{aligned}
\tag{2}
$$

with $\Delta\rho = \rho_\perp - \rho_\parallel$ the amplitude of the PHE ; $\rho_\perp = 1/\sigma_D$ constant in magnetic field ; and $\Delta\rho \propto B^2$ (in the type-I case). Note that the sign convention for $\Delta\rho$ from Ref. [2] is opposite that of $\Delta\sigma$ from Ref.[3].

While Equation 2 is the equation generally adopted in the community, we note that it is only valid in the absence of anisotropies in the zero-field conductivity matrix, i.e. when the Drude conductivity doesn't depend on the direction of the current ($\sigma_D(I_x) = \sigma_D(I_y)$). Even in the absence of anisotropies, the quadratic field dependence of $\Delta\rho$ is only correct in the small oscillations limit, i.e. when $r = \Delta\sigma/\sigma_D << 1$. As this ratio increases (i.e. as the field increases), the field dependence of $\Delta\rho$ slows down, and it ultimately saturates at high field, with

$$
\lim_{r \to \infty} \Delta\rho(r) = 1/\sigma_D.
$$

Together with the mixed linear and quadratic field dependence terms expected in the case of tilted WCs, this may explain why most experimental papers studying the PHE in topological semi-metals have reported sub-quadratic field dependences [4, 15–17]. The expected angular dependence of the resistance is represented in Figure 1.a, with $R_{xx}$ in blue and $R_{yx}$ in red. The longitudinal resistance $R_{xx}$ is maximal when magnetic field and current are aligned, while the transverse resistance $R_{yx}$ vanishes in this configuration.

### Planar Hall effect in nanostructures of t-PtBi$_2$

We studied the planar Hall effect in three nanostructures of PtBi$_2$, obtained by mechanical exfoliation from

high-quality single crystals of $PtBi_2$ grown using the self-flux method [18]. The flakes produced by the exfoliation have typical widths exceeding 10 $\mu$m and thickness ranging from a few dozen to a few hundred nanometers. The flakes were contacted with Cr/Au using standard e-beam lithography techniques. Prior to the metal deposition, a small Ar-etch was performed to eliminate any surface oxidation. The main samples used in this study are denoted as $D1$ (70 nm thick), $D2$ (126 nm thick) and $D3$ (41 nm thick), and their measurement configurations are shown in Figure 2. In previous studies, the two-dimensional superconductivity of these samples was studied in details at sub-Kelvin temperatures [8], and an anomalous planar Hall effect was reported for $D1$ and $D2$ [19]. Here, we focus on measurements performed from room temperature down to 1 K, above the superconducting transition. No evidence of aging effects was observed between our studies, as indicated by the absence of any measurable change in the residual resistance ratio ($RRR = R(300K)/R(4K)$, see supplementary information Ref. [19]).

PHE in $PtBi_2$ was measured first in 2-dimensional in-plane magnetic field mappings of the longitudinal and transverse resistance at 1 K with a 3D vector magnet (Figure 2). The resistance was measured using external lock-in amplifiers, with an AC current of 20 $\mu$A at a frequency of 113 Hz, with an integration time of 300 ms. Although the magnetic field range is limited to 1.5 T in either in-plane directions in this setup, the AMR and PHE are clearly visible in the mappings, through their $\pi$-periodicity and $\pi/4$ rotation between longitudinal and transverse measurements. The features at low field in Figure 2.g,h are associated with remnants of the superconductivity in the sample. We extract from the mappings the angular dependence of the resistance at fixed fields $B_0$, between 500 mT and 1.5 T. In order to have enough data points for analysis, we consider all points within 20 mT of $B_0$ (i.e. we extract all points with $|B - B_0| < 20$ mT).

The angular dependence obtained at 1.5 T and 1 K is shown for all three samples in Figure 2.j,k,l, and displays the features expected for the PHE for both longitudinal ($R_{xx}$, top panels) and transverse ($R_{yx}$, bottom panels) resistances. The maxima of longitudinal resistance correlate well with the expected orientation of the current in the samples. The data also shows some visible $2\pi$-periodic signal, which may come from stray out-of-plane-field magnetoresistance (MR) due to a misalignment between the samples' planes and the magnetic field plane. We can fit the data very well using a constrained PHE model based on Equation 2, which takes into account a $2\pi$-periodic contribution (more details on this model in the next sections, see Equation 4).

**Field and temperature dependences of the PHE**

To study the PHE in more details, we measured samples $D1$ and $D2$ in a Dynacool 14T PPMS using an insert equipped with a mechanical 2D rotator. By rotating the sample with the rotator, the angle $\varphi$ between the fixed-axis magnetic field and the applied current can be adjusted over a full range of 360°. The resistance was measured using external lock-in amplifiers, with an AC current of 100 $\mu$A at a frequency of 927.7 Hz, with an integration time of 300 ms. In the main text, we will focus on results obtained for sample $D1$, although the same analysis was done for sample $D2$, with similar conclusions (see Supplementary Materials).

For measurements done at $T = 5$ K and $B = 1, 2, 3, 4, 5, 6, 7, 10$ T, as well as at $B = 14$ T and $T = 5, 10, 20, 50, 300$ K, 10 points were measured in succession at each angular position, and their resistance was averaged. The angular step for each measurement was 1°. More precise measurements were taken at $B = 14$ T and $T = 5, 100, 200$ K, with 40 points measured in succession at each angular position, and their resistance averaged. The angular step for each measurement was 0.5°, and the results were interpolated with a step of 1°, to perform the analysis in the same way for each pair of (B,T) parameters.

The measurements done at $T = 5$ K and $B = 14$ T are presented in Figure 3.a, and show a large $\pi$-periodic oscillation corresponding to the PHE. Again, the phase of the oscillations correlates well with the expected orientation of the current in the sample. The PHE is already visible at 1 T, the lowest magnetic fields measured (see Figure 4.a), and its magnitude increases with the magnetic field, with a power law that remains in very good agreements with the low field data (see Supplementary Materials). The amplitude of the PHE decreases with increasing temperature (Figure 4.b), and the PHE is very robust with temperature, and can be observed at 14 T up to room temperatures, as shown in Figure 3.c.

**Analysis of the results**

Beyond the $\pi$-periodicity and $\pi/4$ offset between the longitudinal and transverse oscillations, and the the correlation between the orientation of the current and the phase of the oscillations, an important characteristics of the PHE is the expected equal amplitude of both oscillations in resistivity (see Equation 2). A more complex analysis is required to confirm this point in real systems, as measurements can only access the resistance, and not the resistivity. In the ideal case, the conversion between the two follows the formulae:

$$R_{xx} = \frac{L}{W \times t} \cdot \rho_{xx} = \frac{N_\square}{t} \cdot \rho_{xx},$$
$$R_{yx} = \frac{1}{t} \cdot \rho_{yx}, \tag{3}$$

with $L$ the distance between the longitudinal contacts, $W$ the width of the sample, $t$ its thickness, and $N_\square = L/W$ the number of squares between the longitudinal contacts. In this paper, and unless stated otherwise, we do not consider the thickness of the sample in this calculation (i.e. we take $t = 1$) as it doesn't change the relative amplitudes between $R_{xx}$ and $R_{yx}$.

A careful analysis of the measurements is necessary, as we need to take into consideration the contributions of several additional signals, coming from different origins. First, as the shape of the samples deviates from the traditional Hall-bar, geometrically estimating accurately $N_\square$ between a pair of contacts is not trivial. This is further complicated by the position of the contacts on top of the flake, going inwards. It has been shown that such intrusive contacts can distort current flow and significantly reduce the measured amplitude of the transverse signal (possibly by as much as 50-75% in geometries similar to ours) [20]. We therefore consider an arbitrary reduction in the amplitude of the transverse resistance measured compared to its full amplitude.

Second, if the transverse contacts are not perfectly aligned orthogonally to the direction of the current, the resistance measured between them will include a longitudinal contribution, which can be significant as the longitudinal and transverse signals in the PHE have the same amplitude. In a traditional Hall configuration, such extra contributions due to contact misalignment can be removed by symmetrising (resp. asymmetrising) the longitudinal (resp. transverse) resistance in magnetic field, as the longitudinal and Hall signals are expected to be respectively even and odd in field. However in our system and configuration, such a procedure cannot be applied, such that we must consider an additional longitudinal contribution to the transverse resistance, with an arbitrary amplitude, in the fit.

Finally, as mentioned above, we must consider that the sample may not lie exactly in the rotation plane. When a magnetic field is applied in the rotation plane, this will result in a component of the magnetic field being perpendicular to the sample's plane, and thus in a regular magnetoresistance component in both longitudinal and transverse resistances. As the sample is rotated, the amplitude of the out-of-plane field will vary $2\pi$-periodically, resulting in additional magnetoresistance contributions to the longitudinal and transverse resistances (with $\pi$- and $2\pi$-periodicity, respectively). We consider additional $2\pi$-periodic signals in both resistances, to account for any background.

When all these contributions are considered, our system can be described with

$$R_{xx}(B, \varphi) = \mathrm{R}_{xx}^{2\pi}(B, \varphi) + N_\square \cdot \rho_{xx}^{\mathrm{PHE}}(B, \varphi),$$
$$R_{yx}(B, \varphi) = \mathrm{R}_{yx}^{2\pi}(B, \varphi) + C_T \cdot \rho_{yx}^{\mathrm{PHE}}(B, \varphi) + C_L \cdot R_{xx}(B, \varphi), \tag{4}$$

with

$$\rho_{xx}^{\mathrm{PHE}}(\varphi) = \rho_\perp - \Delta\rho \cos^2 \varphi,$$
$$\rho_{yx}^{\mathrm{PHE}}(\varphi) = -\Delta\rho \cos \varphi \sin \varphi,$$
$$\mathrm{R}_{xx}^{2\pi}(B, \varphi) = A_{xx}(B) \cos(\varphi - \varphi_L),$$
$$\mathrm{R}_{yx}^{2\pi}(B, \varphi) = A_{yx}(B) \cos(\varphi - \varphi_T) + C. \tag{5}$$

Here, $\Delta\rho = \rho_\parallel - \rho_\perp$ is the amplitude of the PHE ; $\varphi = \widetilde{\varphi} - \varphi_{\mathrm{PHE}}$ is the angle between the magnetic field and the current in the sample, with $\widetilde{\varphi}$ the angular position of the sample set by the rotator and $\varphi_{\mathrm{PHE}}$ the rotator angle for which current and field are aligned ; $\mathrm{R}_{xx}^{2\pi}$ and $\mathrm{R}_{yx}^{2\pi}$ are $2\pi$-periodic background contributions with arbitrary angular origins $\varphi_L$ and $\varphi_T$ ; $C$ is a field-independent offset of the transverse resistance ; and $N_\square$, $C_T$ and $C_L$ are respectively the effective number of squares between longitudinal contacts, the correcting factor for the transverse resistance due to the invasive contacts, and the correcting factor due to the misalignment of the contacts. The same analysis is performed in temperature, with the addition of a temperature-dependent vertical offset to the background terms to account for the resistance increasing with temperature:

$$\mathrm{R}_{xx}^{2\pi}(T, \varphi) = A_{xx}(T) \cos(\varphi - \varphi_L) + C_{xx}(T),$$
$$\mathrm{R}_{yx}^{2\pi}(T, \varphi) = A_{yx}(T) \cos(\varphi - \varphi_T) + C_{yx}(T). \tag{6}$$

Due to the large number of unknown parameters, it is not possible to get meaningful values for the different variables by fitting a single set of angular dependence (i.e. $R_{xx}(B_0, \varphi)$ and $R_{yx}(B_0, \varphi)$) with Equation 4. We can however overcome this issue by noting that most of these parameters are geometric and therefore independent of the external magnetic field. Thus, by fitting $R_{xx}(B, \varphi)$ and $R_{yx}(B, \varphi)$ together at multiple fields B, and fixing the geometric parameters as global parameters across all fits, we can extract meaningful values for each parameter, and recover the amplitude of the PHE in resistivity.

The results obtained are shown as thick lines in Figure 3, and present an excellent agreement with the experimental data for both samples, with only small deviations from the model at low fields and high temperatures, which we will discuss shortly (see Figure 5). The dependence of the PHE amplitude $\Delta\rho$ with field and temperature is shown in Figure 4. $\Delta\rho$ increases with field, following the power laws $\Delta\rho \propto B^{1.24}$, which is consistent with low field measurements (see Supplementary materials), and is notably lower than the expected quadratic behavior expected for the pure chiral anomaly effect [2].

As mentioned before, both BC effects and orbital magnetic moment effects could contribute to this sub-quadratic field dependence [21]. Importantly, $\Delta\rho$ starts decreasing with temperature above $T \sim 20$ K, and remains clearly visible up to room temperature, with $\Delta\rho \simeq 0.34\ \mu\Omega$.cm, which is relatively large for a non-magnetic system.

## DEVIATIONS FROM PHE MODEL

As stated above, and as can be seen in Figure 3.c, the constrained model of Equation 4 deviates from the data towards low magnetic fields and high temperatures. In the following, we will detail and analyse these deviations and provide possible explanations.

### Planar Hall effect at low field and at high temperature

At low temperature $T = 5$ K and low magnetic fields $B \leq 3$ T, while the oscillations from the PHE are still clearly visible in both samples (see Figure 5.a,b,c), the constrained model (thick lines) deviates significantly from the data, due mainly to an phase offset of the oscillations as well as a vertical offset. The amplitude of the oscillations is however well represented by the model. Both offsets decrease quickly with field. At high field ($B = 14$ T) and high temperature $T \geq 100$ K on the other hand, there are no vertical or phase offsets between the constrained model and the data. However, while the model still fits closely the data in $R_{yx}$, it starts deviating at high temperatures from the data in $R_{xx}$, as shown in Figure 5.e,f,g (thick lines). This is due to both oscillations no longer sharing the same amplitude, which is incompatible with the PHE model. In order for the fit to keep converging meaningfully at high temperature, we chose to artificially give a higher weight to the data in $R_{yx}$ than in $R_{xx}$, which is why the deviations is seen only in $R_{xx}$.

### Simple model

While the data deviates from the constrained model of Equation 4, we note that the general shape of the data can still accurately be described by $\pi$-periodic oscillations. Thus, to study the deviations from the constrained model in more details, we fitted the data for $R_{xx}$ and $R_{yx}$ independently with an unconstrained model, which includes only two contributions: $\pi$-periodic, and $2\pi$-periodic:

$$R(\varphi) = C + A^{2\pi} \cdot \cos\left(\varphi - \varphi_{2\pi}\right) + A^{\pi} \cdot \cos\left[2(\varphi - \varphi_{\pi})\right], \tag{7}$$

with $C$ an angle-independent constant, and $A^{\pi}$ and $A^{2\pi}$ the amplitudes of the $\pi$- and $2\pi$-periodic signals, with angular origins $\varphi_{\pi}$ and $\varphi_{2\pi}$, respectively. The results are shown in Figure 5 in dashed lines, and fit the data closely at all fields and temperatures.

The phase and amplitude of the $\pi$-periodic oscillations in $R_{xx}$ and $R_{yx}$ can be extracted from this model, and the field and temperature dependence are shown in Figure 5.d,h. For comparison, we show the amplitudes reduced by their values at $B = 14$ T, $T = 5$ K. We also subtract 45° from the phase in $R_{yx}$, to account for the expected $\pi/4$ shift between $R_{yx}$ and $R_{xx}$. As expected, the amplitudes of the oscillations in $R_{xx}$ and $R_{yx}$ have the same field dependence at low temperature, in both samples (see Figure 5.d, top panel). The phase of the oscillations, however, shows a strong variation at low magnetic field (Figure 5.d, bottom panel). While the phase of the oscillations in $R_{yx}$ stabilizes quickly, at about 2 T, it continues increasing slowly in $R_{xx}$ over the full angular range. The difference in phase for both oscillations eventually come within a few degrees of the expected $\pi/4$ at higher field, without reaching that value. The other sample, $D2$, shows slightly different features (see Supplementary materials), with only a small variability of the phase at low field, which might be attributed to a lower signal-to-noise ratio. The shift between the oscillations in $R_{xx}$ and $R_{yx}$ deviates from the expected value of 45° by about 5°, which remains about constant over the entire magnetic field range. The variability of the phase in $D1$ might be related to current jetting in the sample, which does not have a standard Hall bar shape: If the current lines change orientation as the magnetic field is increased, e.g. to minimize the magnetoresistance, this could lead to a change in the phase of the PHE, which depends on the relative orientation of the magnetic field and the current. As $D2$ is closer to the Hall bar shape, the effect of current jetting would be expected to be lower, and the phase variation in field attenuated as well. The deviation from the $\pi/4$ offset between $R_{xx}$ and $R_{yx}$ might also be related to the shape of the samples: As neither $D1$ nor $D2$ are perfect Hall bars, the resistance between the transverse contacts will also include a longitudinal contribution. Since $a \cdot \cos\phi + b \cdot \sin\phi = R \cdot \cos(\phi - \alpha)$, with $R = \sqrt{a^2 + b^2}$ and $\alpha = \tan^{-1}(b/a)$, the longitudinal contribution is equivalent to a renormalization and phase offsetting of the transverse signal.

The same analysis is performed at $B = 14$ T and temperatures ranging from $T = 5$ K to $T = 300$ K, and the results are shown in Figure 5.h. As expected, no large variation is observed in the phase of the oscillations, with a few degrees separating the phase in $R_{xx}$ and $R_{yx}$, as before. However, the temperature dependence of the amplitude of the oscillations is now different between the two resistances, with the amplitude in $R_{xx}$ decreasing more slowly than in $R_{yx}$. This effect is clearly visible in both samples (see Supplementary materials for $D2$), and is the

reason for the deviation between the constrained model and the data.

There are, broadly, two mechanisms which could cause this discrepancy between $R_{xx}$ and $R_{yx}$. The first mechanism relates to the geometry of the sample and the flow of current. In our analysis, we have considered these parameters (i.e. $N_\square, C_T, C_L$ etc.) as independent of magnetic field and temperature. As these parameters relate the resistance to the resistivity, they provide us directly with the link between the amplitude of the oscillations between $R_{xx}$ and $R_{yx}$. However, if (some of) these parameters were to vary with temperature, this would change the expected balance between the two amplitudes, and may cause the observed discrepancy. Although the exact geometry of the current flow may vary slightly with temperature due to inhomogeneities in the sample, for instance around the invasive contacts, leading to slightly different temperature dependence of the resistance, we cannot think of an effect strong enough to cause the observed difference in amplitude. The second mechanism would be the existence of a second $\pi$-periodic effect in $R_{xx}$, distinct from the PHE. This effect would need to have a small amplitude at low temperature with respect to the PHE, as the latter accounts well for the data at low temperature. If such an effect did exist, and its amplitude decreased more slowly than that of the PHE, then the balance between this effect and the PHE would change as the temperature increases, with the PHE becoming less predominant at higher temperature, leading to the extra signal observed. One possibility for such a signal would be a contribution to the magnetoresistance from a small out-of-plane magnetic field $B_\perp$, due to a misalignment between the sample's plane and the rotation plane of the rotator. Such a misalignment would result in a $2\pi$-periodic variation of $B_\perp$ around $B_\perp = 0$, as the sample is rotated. This would result in a $2\pi$-periodic contribution to $R_{yx}$, as the Hall resistance is odd in out-of-plane field, but would give a $\pi$-periodic contribution in $R_{xx}$, as the longitudinal resistance is even in out-of-plane field. A tilt of the sample with respect to the rotation plane would therefore result in an additional $\pi$-periodic contribution in $R_{xx}$, with no such additional contribution in $R_{xx}$. Depending on the axis of this tilt with respect to the orientation of the current in the sample, this additional contribution would either add up to or cancel out some of the oscillation due to the PHE. We note that a misalignment of the rotation plane with respect to the magnetic field axis would result in a constant out-of-plane field, independent on the angular position of the sample, and would therefore not have an influence on the $\pi$-periodic signal. As the PHE and the longitudinal magnetoresistance (LMR) are not expected to fully share a common physical origin in our material , they have no reason to share a similar temperature dependence. It is therefore possible that the LMR has a slower decrease in temperature than the PHE, and could account for the additional oscillation seen in our measurements.

## DISCUSSION

As we reported previously, t-PtBi$_2$ was predicted to be a Weyl semimetal from DFT calculations [8, 19], and several direct experimental confirmations have since been found in ARPES [10] and STM [11], where Fermi arcs have been observed. Recently, we also found evidence of an anomalous planar Hall effect (APHE) in PtBi$_2$, attributed to the magnetic field-conversion of topological nodal lines into Weyl nodes [19]. Outside of the PHE (and APHE), no transport signature of the topology of PtBi$_2$ has been reported as of yet.

The most well known transport signature of topological semimetals is the negative longitudinal magnetoresistance, which is associated with the chiral anomaly [22]. However, it has been recently understood that in Weyl semimetals effects of orbital magnetic moments could result both in a positive longitudinal magnetoresistance [23] and in a positive transversal magnetoresistance [21] when considering tilted Weyl cones. The anisotropy in these responses is expected to lead to a PHE that therefore represents one of the main transport signature in topological semimetals. It is important to point out that the observation of a PHE in non-magnetic materials does not necessarily implies the presence of topological degeneracies. The Lorentz-force induced orbital magnetoresistance [24] is also expected to be anisotropic due to cancellation of Lorentz force with collinear electric and magnetic fields and can therefore yield a PHE. In this regard, it is important to note that a very large magnetoresistance (with out-of-plane magnetic field) has been reported in PtBi$_2$ [8]. We can thus expect the orbital magnetoresistance to be significant, and therefore its anisotropies could contribute to a PHE. Nonetheless, our measurements are consistent with the predictions of Weyl physics in PtBi$_2$, and furthermore all our results can be understood in that context, without needing to invoke additional effects. The observation of a PHE in PtBi$_2$ therefore reinforces with charge-transport previous observations from spectroscopy techniques [10, 11] as to its Weyl nature.

As mentioned above, we have recently predicted PtBi$_2$ to be a nodal line semimetal, and the nodal lines are expected to convert into multiple Weyl nodes under infinitesimal magnetic field [19]. These field-created nodes are both of type-I and type-II, and are expected to appear at various energies. While it is difficult to distinguish the contributions of each Weyl node to the PHE, we note that we cannot dismiss the contribution of the additional Weyl nodes simply due to their distance to the Fermi energy. Furthermore, as the $k$-space distance between these Weyl nodes is unusually large (by the very nature of their creation mechanism), the PHE originating from these Weyl

nodes might actually be stronger than the one originating from the zero-field 12 Weyl nodes. Indeed, the contribution of the chiral anomaly to the PHE comes from the accumulation of carriers around one Weyl cone with a given chirality, and a corresponding depletion of carriers around a Weyl cone with the opposite chirality [22]. When the Fermi energy exceeds the Lifshitz-transition energy, inter-cone scattering will tend to reduce this inbalance in chiral carriers. With WN close in $k$-space, such inter-cone scattering will occur via long-range disorder, and may therefore be very efficient. However when the WN are far in $k$-space, the main mechanism behind inter-cone scattering will be due to short-range disorder, which might be far less efficient. Therefore, we cannot exclude that the large, and very robust, PHE we observe might be due, in part, to the many Weyl nodes originating from the nodal lines. This would help explain the sub-quatradicity of the field dependence of the PHE, if both type-I and type-II WNs contribute to the PHE, as the latter will partially contribute a linear field-dependence.

## CONCLUSION

In this paper, we presented the first measurement of the planar Hall effect in nanostructure of the van-der-Waals layered non-magnetic type-I Weyl semimetal trigonal-$PtBi_2$, and study their dependence in magnetic field and temperature. The discovery of a PHE in $PtBi_2$ is significant, as it is an expected signature of Weyl semimetals. We have found that the PHE is present already at magnetic fields as low as 1 T, and is robust up to room temperature. We also unveiled some deviations from the theoretical expectations for the PHE in Weyl semimetals, with a sub-quadratic field dependence of the amplitude, which may originate from a combination of the Weyl-cones' tilt (e.g. from some type-II Weyl nodes originating from the nodal lines) and a deviation from the simple sine-wave model when the amplitude of the oscillations is large.

Overall, this study reinforces our understanding of the quantum geometry of trigonal-$PtBi_2$, which is of particular interest in the context of the recent discovery of superconductivity in this material, with two-dimensional superconductivity reported in bulk nanostructures, and surface Fermi-arc supported superconductivity seen in ARPES.

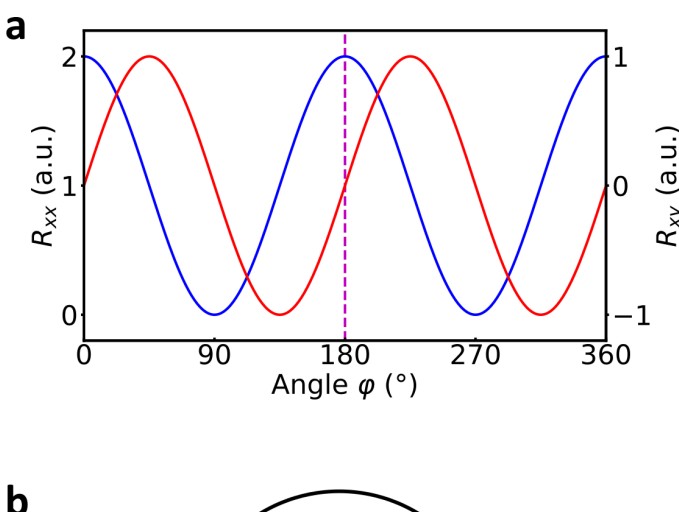

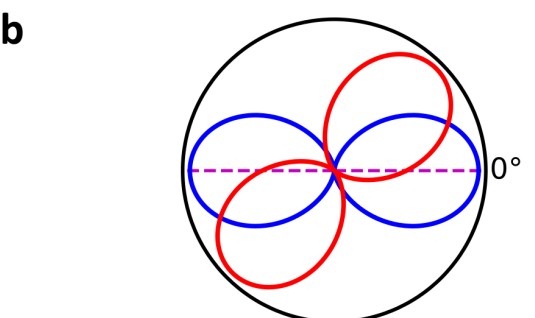

FIG. 1. **a,b**: Typical angular dependence of the conventional planar Hall effect, in Cartesian (**a**) and polar (**b**) coordinates. Both the longitudinal (anisotropic magnetoresistance, $R_{xx}$, blue) and transverse (planar Hall effect, $R_{yx}$, red) resistances exhibit a $\pi$-periodic angular dependence, with a $\pi/4$-offset between them. The origin of the oscillation is set by the direction of the electric field (current).

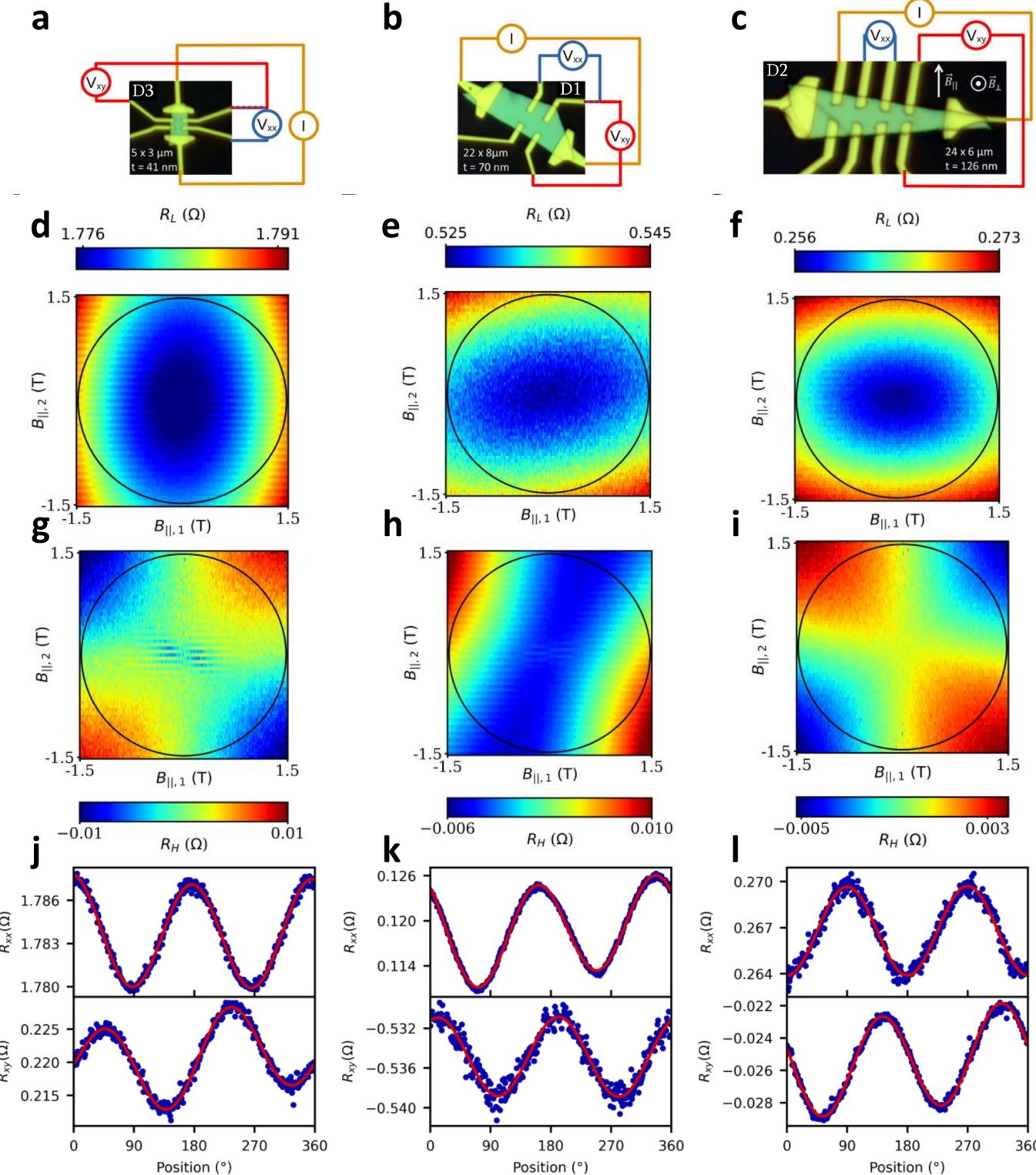

FIG. 2. **a,b,c**: Optical pictures and sample configurations for D3 (41 nm), D1 (70 nm) and D2 (126 nm). The measurements related to each sample is shown below it. **d-i**: In-plane magnetic field $(B_y - B_z)$ mappings of the longitudinal ($R_{xx}$, d,e,f) and transverse ($R_{yx}$, g,h,i) resistances. All mappings were measured simultaneously by sweeping $B_{||,1}$ at fixed $B_{||,2}$, and increasing $B_{||,2}$ in steps of 50 mT. All mappings show the expected four-fold symmetry expected for the PHE, with sample-orientation-dependent phase and $\pi/4$ shift between longitudinal and transverse configurations. **j-l**: Angular dependence of $R_{xx}$ (top panels) and $R_{yx}$ (bottom panels) extracted from the mappings **d-i**, for a field $B = 1.5$ T corresponding to the black circles. The data is well fitted by the PHE model (in red, see Equation 4). The phase of the oscillations in each sample can be correlated to the presumed current orientation in the sample.

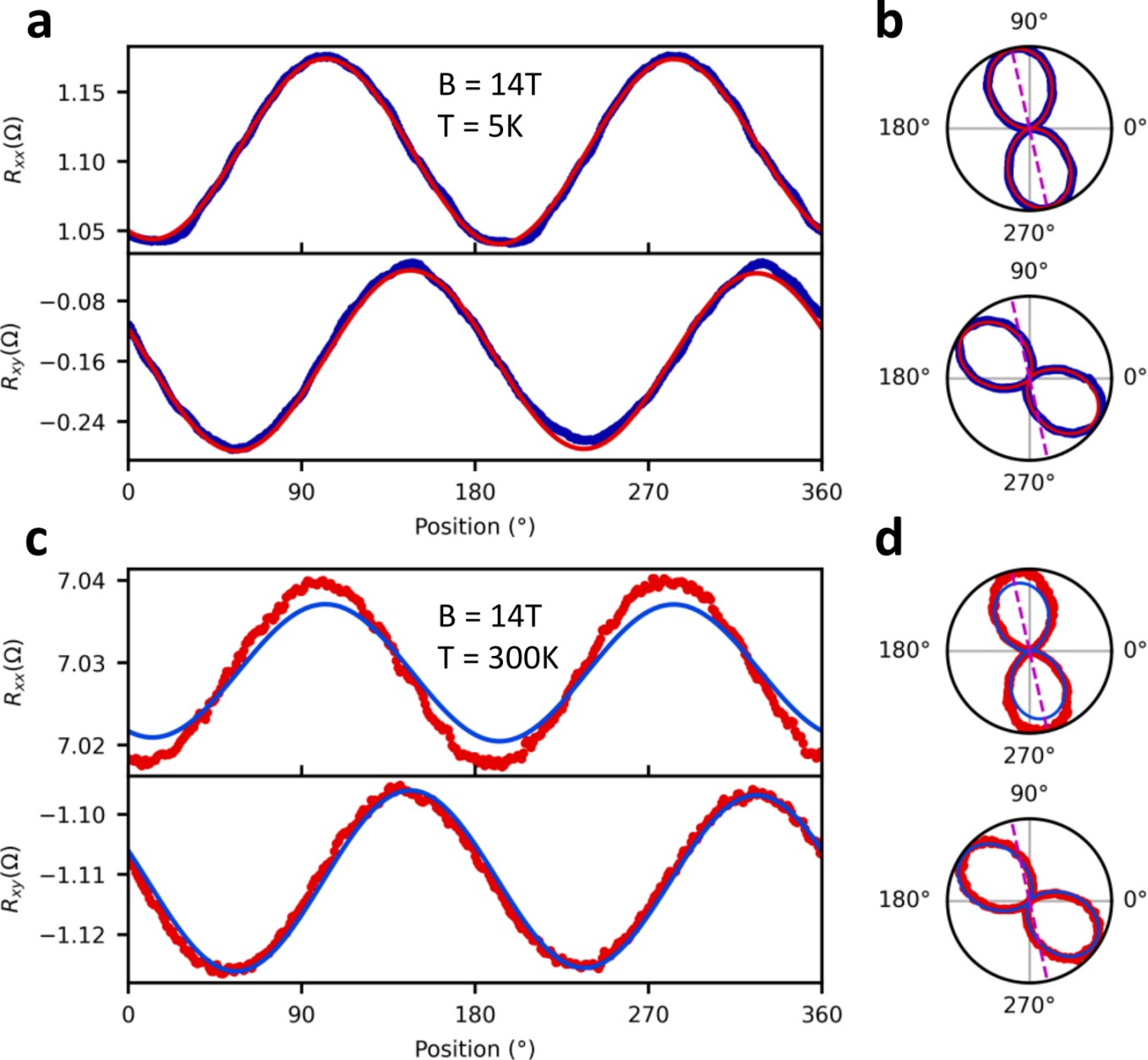

FIG. 3. **a,c**: Angular dependence at 14 T and 5 K (a) or 300 K (c) of the longitudinal ($R_{xx}$, top panels) and transverse ($R_{yx}$, bottom panels) resistances for sample D1, in Cartesian coordinates. Fits to the PHE model (Equation 4 are shown in red (a) and blue (c). **b,d**: Same data as in **a,c**, in polar coordinates. The dashed-line represents the orientation of the current estimated from the data.

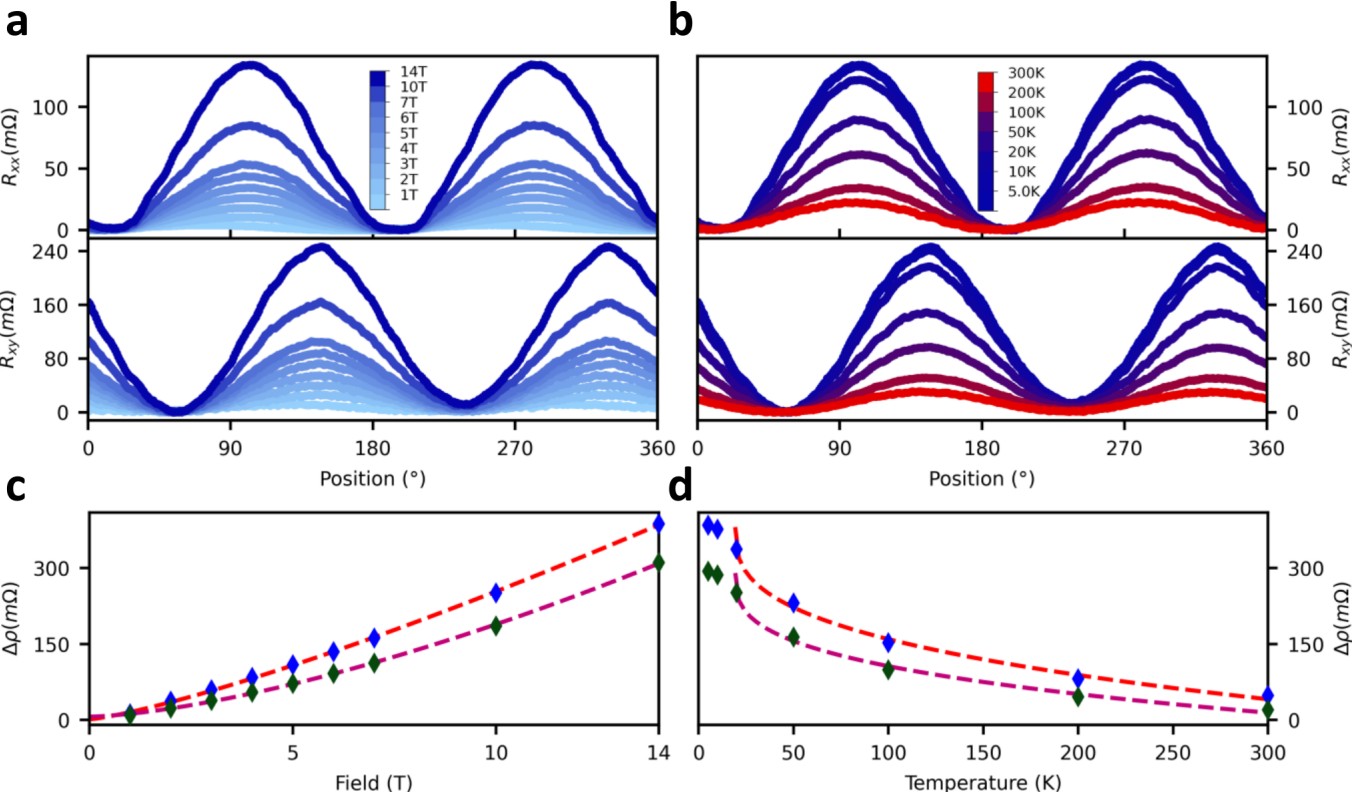

FIG. 4. **a,b**: Angular dependence of the longitudinal ($R_{xx}$, top panels) and transverse ($R_{yx}$, bottom panels) resistances of samples D1 at 5K and multiple fields from 1T to 14T (a) and at 14T and multiple temperatures from 5K to 300K (b). The plots are shifted vertically for visibility, to share a minimum at 0 Ω. **c,d**: Field (c) and temperature (d) dependence of the PHE amplitude $\Delta\rho$ extracted from fits of the data in (a,b) with Equation 4 for samples D1 (blue diamonds and dashed-red line, respectively), and for sample D2 (green diamonds and dashed-magenta line, respectively, see SM). The field dependence of $\Delta\rho$ is well fitted with a sub-quadratic power law for both samples. The temperature dependence of $\Delta\rho$ cannot be fitted with an exponential decay law, and is fitted with a power law.

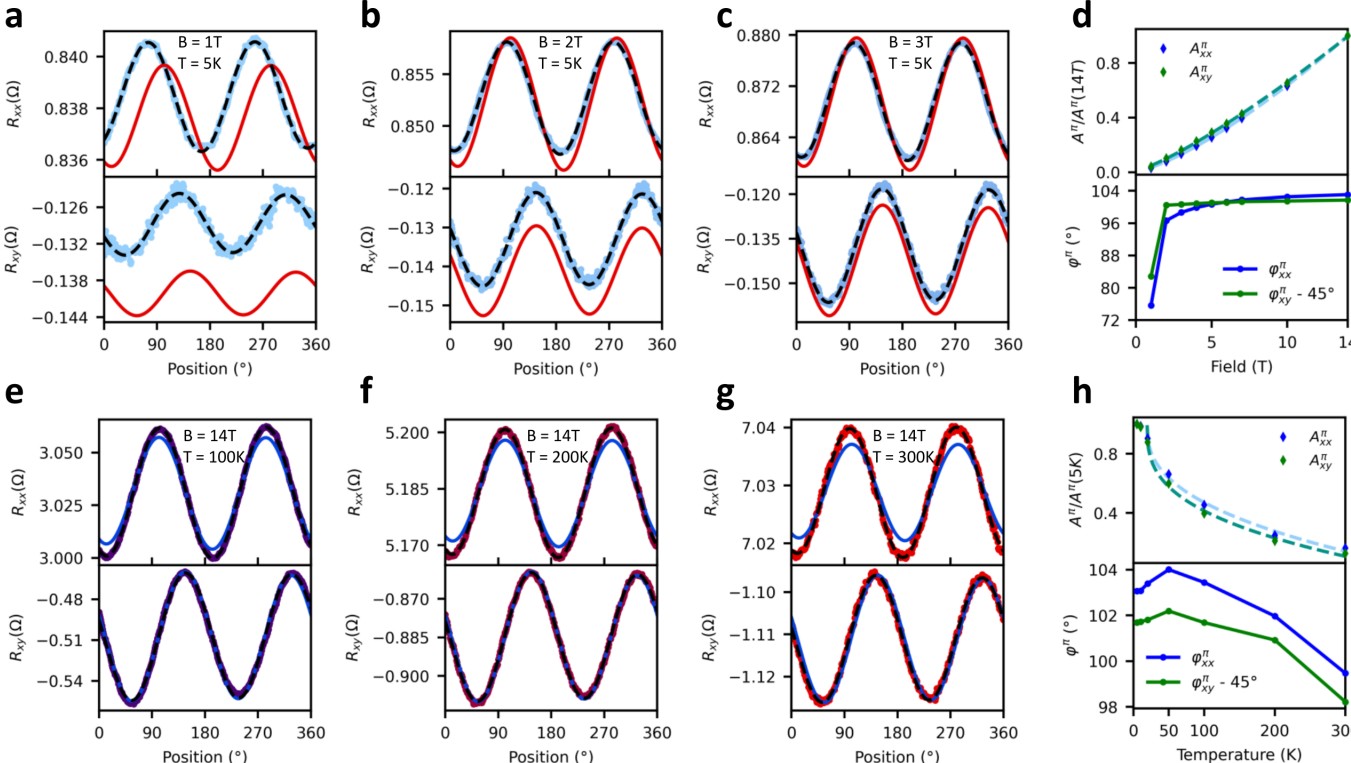

FIG. 5. **a,b,c**: Angular dependence of the longitudinal ($R_{xx}$, top panels) and transverse ($R_{yx}$, bottom panels) resistances at 5 K and 1T (a), 2T (b) and 3T (c). As the result of dephasing and unaccounted-for offset, the constrained fits of the PHE model (Equation 4, in red) start deviating from the measurements at low fields. The data can still be fitted very well with an unconstrained $\pi$-periodic fit (in dashed black). **d**: Field dependence of the amplitude (top panel, renormalized to its value at 14T) and phase (bottom panel) of the unconstrained fits of the longitudinal ($R_{xx}$, blue) and transverse ($R_{yx}$, green) resistances. The renormalized amplitudes of the oscillations in $R_{xx}$ and $R_{yx}$ have the same field dependence, as expected for the PHE, however the phase of the oscillations can change significantly with the magnetic field. **e,f,g**: Angular dependence of the longitudinal ($R_{xx}$, top panels) and transverse ($R_{yx}$, bottom panels) resistances at 14 T and 100 K (e), 200 K (f) and 300 K (g). The constrained fits of the PHE model (Equation 4, in blue) start deviating from the measurements in $R_{xx}$ at high temperature, as the amplitude of the oscillations are smaller than anticipated from those in $R_{yx}$. The data can still be fitted very well with an unconstrained $\pi$-periodic fit (in dashed black). **h**: Temperature dependence of the amplitude (top panel, renormalized to its value at 5 K) and phase (bottom panel) of the unconstrained fits of the longitudinal ($R_{xx}$, blue) and transverse ($R_{yx}$, green) resistances. The renormalized amplitudes of the oscillations in $R_{xx}$ and $R_{yx}$ have a different dependence in temperature, as the amplitude in $R_{xx}$ decreases more slowly than that in $R_{yx}$. The phase of the oscillations changes slightly with the temperature.

* j.dufouleur@ifw-dresden.de

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

# Planar Hall effect in nanostructures of trigonal-PtBi$_2$: Supplementary Materials

Arthur Veyrat,[1,2,3] Klaus Koepernik,[1,2] Louis Veyrat,[1,2,4] Grigory Shipunov,[1,2] Saicharan Aswartham,[1,2] Jiang Qu,[1,2] Ankit Kumar,[1,2] Michele Ceccardi,[5,6] Federico Caglieris,[6] Nicolás Pérez Rodríguez,[1,2] Romain Giraud,[1,2,7] Bernd Büchner,[1,2,8] Jeroen van den Brink,[1,2,8] Carmine Ortix,[9,*] and Joseph Dufouleur[1,2,10,†]

[1]*Leibniz Institute for Solid State and Materials Research (IFW Dresden), Helmholtzstraße 20, D-01069 Dresden, Germany*
[2]*Würzburg-Dresden Cluster of Excellence ct.qmat, Dresden, Germany*
[3]*Laboratoire de Physique des Solides (LPS Orsay), 510 Rue André Rivière, 91400 Orsay, France*
[4]*CNRS, Laboratoire National des Champs Magnétiques Intenses, Université Grenoble-Alpes,*
*Université Toulouse 3, INSA-Toulouse, EMFL, 31400 Toulouse, France*
[5]*Department of Physics, University of Genoa, 16146 Genoa, Italy*
[6]*CNR-SPIN Institute, 16152 Genoa, Italy*
[7]*Université Grenoble Alpes, CNRS, CEA, Grenoble-INP, Spintec, F-38000 Grenoble, France*
[8]*Department of Physics, TU Dresden, D-01062 Dresden, Germany*
[9]*Dipartimento di Fisica "E. R. Caianiello", Universitá di Salerno, IT-84084 Fisciano (SA), Italy*
[10]*Center for Transport and Devices, TU Dresden, D-01069 Dresden, Germany*

## SUPPLEMENTARY MATERIALS

### Resistivity in the planar Hall effect

It is expected from the theory [1] that the planar Hall effect (PHE) component of the conductivity follows the angular dependence

$$\sigma_{xx}^{\text{PHE}}(\varphi) = \sigma_\perp + \Delta\sigma \cos^2\varphi,$$
$$\sigma_{yx}^{\text{PHE}}(\varphi) = \Delta\sigma \cos\varphi \sin\varphi,$$
(1)

with $\Delta\sigma = \sigma_\parallel - \sigma_\perp$ the amplitude of the oscillation, $\sigma_\perp$ and $\sigma_\parallel$ the conductivity when the in-plane magnetic field is orthogonal or parallel to the electric field (i.e. the current), and $\varphi$ the angle between the magnetic field and the current. When the PHE originates from the chiral anomaly in Weyl semimetals, it is expected that $\sigma_\perp$ is constant in magnetic field, with $\sigma_\perp = \sigma_D$ the drude conductivity, and $\sigma_\parallel = \sigma_D + a \cdot B^2$ increases with magnetic field. In order to obtain the resistivity matrix, we must inverse the conductivity matrix $\sigma = \begin{pmatrix} \sigma_{xx} & \sigma_{xy} \\ \sigma_{yx} & \sigma_{yy} \end{pmatrix}$. From Onsager relations, we have $\sigma_{yx}(B) = \sigma_{xy}(-B)$. Since the magnetic field is in the plane, $\sigma_{xy}(-B, \varphi) = \sigma_{xy}(B, \varphi + \pi) = \sigma_{xy}(B, \varphi)$ as $\sigma_{xy}$ is $\pi$-periodic. Hence, we have $\sigma_{yx}(B) = \sigma_{xy}(B)$. We can also note that $\sigma_{yy}$ can be expressed similarly to $\sigma_{xx}$ by simply exchanging $\sigma_\parallel$ and $\sigma_\perp$: $\sigma_{yy}(\varphi) = \sigma_\parallel - \Delta\sigma \cos^2\varphi$.

We can then invert the conductivity matrixto obtain the resistivity matrix $\rho = \sigma^{-1} = \begin{pmatrix} \rho_{xx} & \rho_{xy} \\ \rho_{yx} & \rho_{yy} \end{pmatrix}$, with

$$\rho_{xx} = \frac{\sigma_{yy}}{\sigma_{xx}.\sigma_{yy} - \sigma_{xy}^2} = \frac{\sigma_{yy}}{\sigma_\perp.\sigma_\parallel},$$
$$\rho_{xy} = \frac{-\sigma_{xy}}{\sigma_{xx}.\sigma_{yy} - \sigma_{xy}^2} = \frac{-\sigma_{xy}}{\sigma_\perp.\sigma_\parallel} = \rho_{yx}$$
(2)

From Equation 2, we recover the shape of the resistivity from [2]:

$$\rho_{xx}(\varphi) = \rho_\perp - \Delta\rho \cos^2\varphi,$$
$$\rho_{yx}(\varphi) = \Delta\rho \cos\varphi \sin\varphi,$$
(3)

with $\Delta\rho = \rho_\perp - \rho_\parallel$, $\rho_\perp = 1/\sigma_\parallel$ and $\rho_\parallel = 1/\sigma_\perp$.

However, while the amplitude of the oscillations in conductivity $\Delta\sigma \propto B^2$ is quadratic in field, this is only true at low field for the resistivity:

$$\Delta\rho = \frac{\sigma_\perp - \sigma_\parallel}{\sigma_\perp \cdot \sigma_\parallel} = -\frac{1}{\sigma_D} \cdot \frac{a/\sigma_D \cdot B^2}{1 + a/\sigma_D \cdot B^2}.$$
(4)

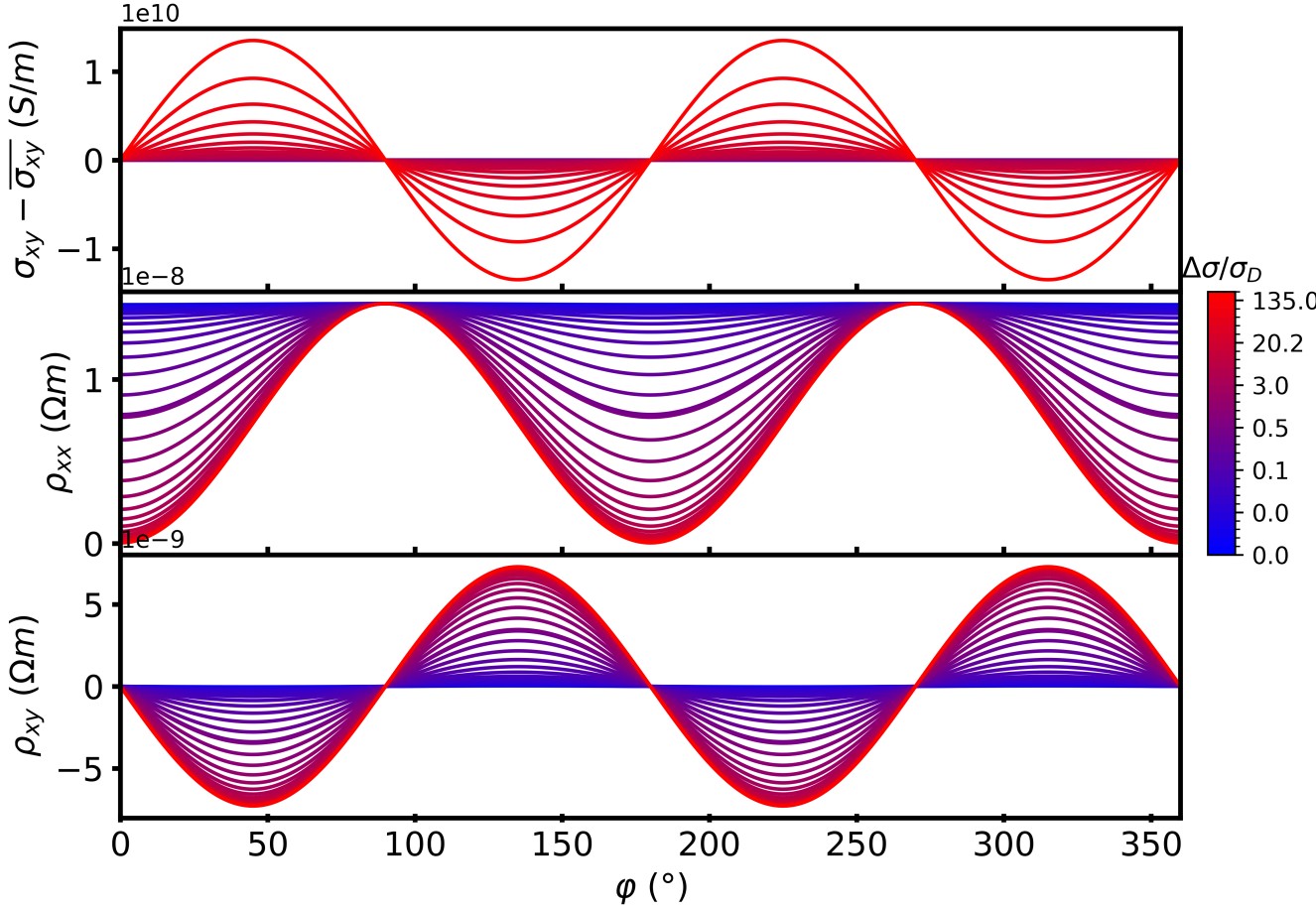

FIG. 1. Simulations showing the angular dependence of the PHE signal in the transverse conductivity (top, vertically offset for visibility), the longitudinal resistivity (middle) and transverse resistivity (bottom), at different magnetic fields from 0.1T (blue) to 300T (red). The corresponding ratio $\Delta\sigma/\sigma_D$ are shown on the right. The parameters used in these simulations are given in section

When $a \cdot B^2/\sigma_D << 1$, we recover the field dependence from [2], $\Delta\rho \propto B^2$, but when $a \cdot B^2/\sigma_D \sim 1$, or alternatively, when $\Delta\sigma \sim \sigma_D$, this approximation breaks down and the oscillations in resistivity begin saturating.

We can estimate at which magnetic field we would expect to see this saturation behaviour in our samples, for instance in $D2$. We can consider that $\sigma_D = 1/\rho_\perp(0T) = 1/R_{xx}(0T) * \dfrac{l}{A}$, with $l$ the distance between the longitudinal contacts, and $A = w * t$, with $w, t$ the width and thickness of the sample, respectively. As a rough estimate, we can take $R_{xx}(0T) \sim 0.836$ $\Omega$, $l \sim 20$ $\mu$m, $w \sim 5$ $\mu$m and $t \sim 70$ nm, which gives $\sigma_D \sim 6.835$ S/m. In order to get $\Delta\rho(14T) = 387$ $m\Omega * \dfrac{l}{A} \sim 6.7e^{-9}$ $\Omega$m, we choose $a = 3e^5$ S/m in the quadratic field-dependence of $\sigma_\parallel$. The field dependence of the PHE predicted with these parameters is shown in Figure 1 and Figure 2 until 300T.

As expected, the oscillations in resistivity remain sinusoidal over the entire field range, and their amplitude saturates at high field. At low field (i.e. $B < 14$ T), we can fit $\Delta\rho$ well with a subquadratic power law: $\Delta\rho(B) \sim b * B^c$, with $b \sim 1.35e^{-10}$ $\Omega$m and $c \sim 1.50$, which is close to the value obtained in our measurements. Although this model is quite simple, as it assumes a simple quadratic field dependence of $\Delta\sigma$ (as in the pure chiral anomaly case in type-I Weyl semimetals), we can see that 14T is close to the inflection point of the field dependence. Even if the field dependence of $\Delta\sigma$ is slightly slower (i.e. subquadratic), measurements in high-field facilities (e.g. up to 60-100T) should show the saturation. One interesting thing to note is that samples with lower resistivities (i.e. higher conductivities) should show a saturation at higher fields than samples with higher resistivities.

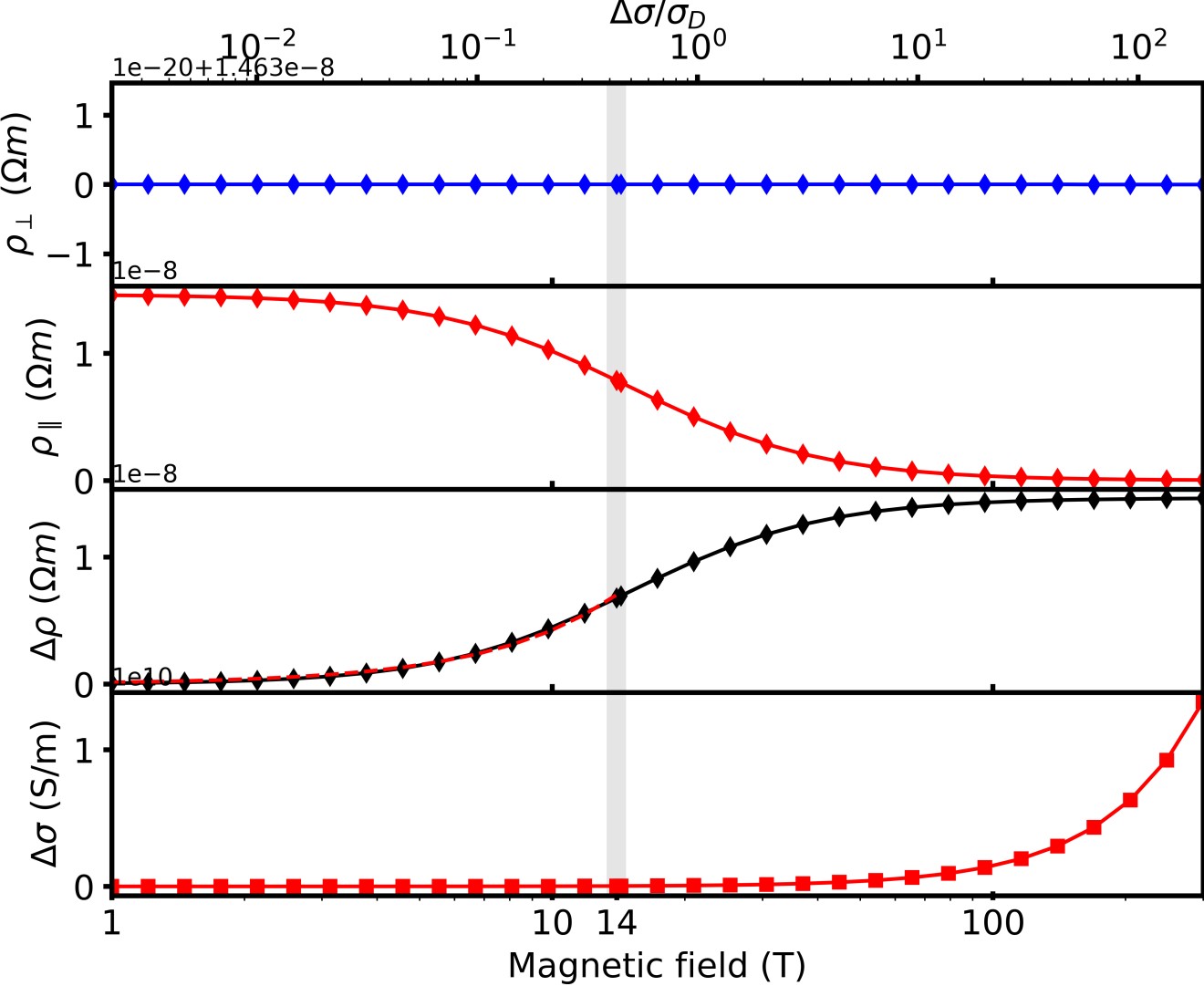

FIG. 2. Field dependence of different parameters of the PHE obtained from fitting the simulations of Figure 1 with Equation 3. The power fit of $\Delta\rho$ is shown in dashed red. The ratio $\Delta\sigma/\sigma_D$ corresponding to the magnetic field is shown at the top. The vertical grey line indicates a magnetic field of 14T.

* cortix@unisa.it

† j.dufouleur@ifw-dresden.de

[1] S. Nandy, G. Sharma, A. Taraphder, and S. Tewari, Physical Review Letters **119**, 1 (2017).

[2] A. A. Burkov, Physical Review B **96**, 041110 (2017).

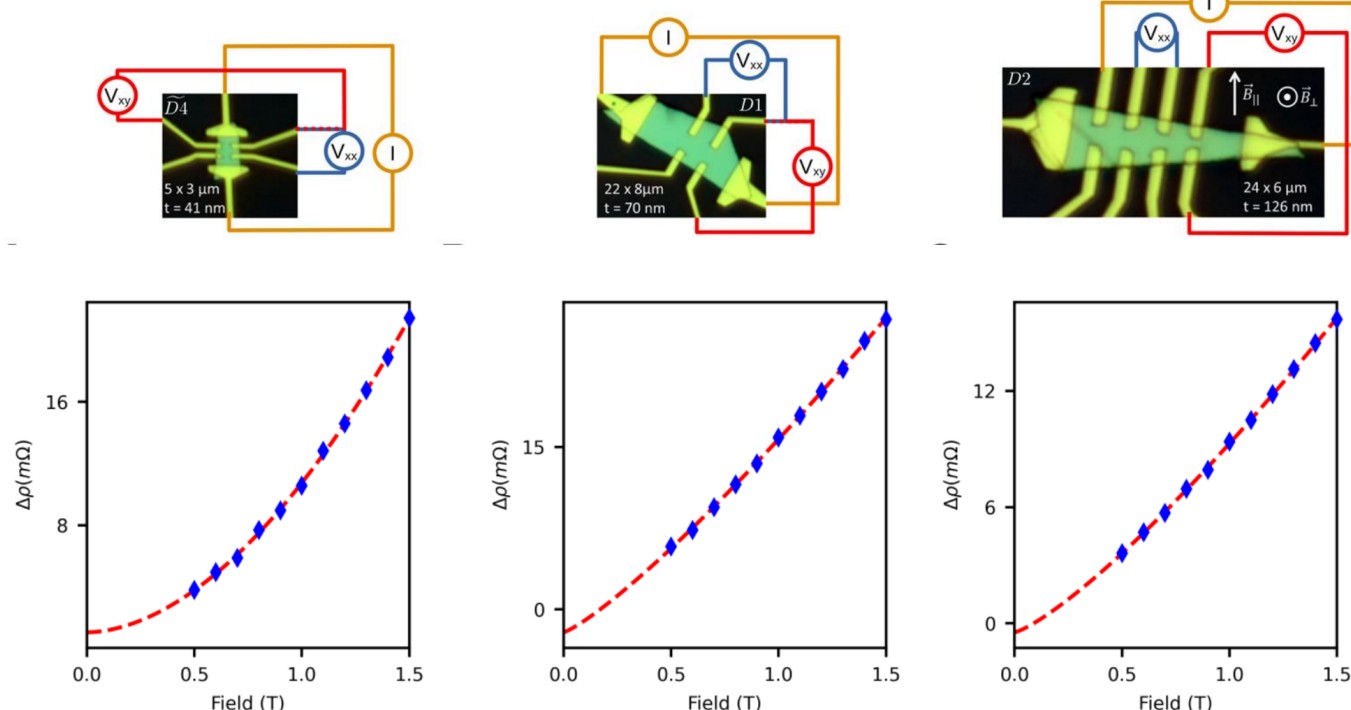

FIG. 3. **a,b,c**: Optical pictures and sample configurations for D3 (41 nm), D1 (70 nm) and D2 (126 nm). The measurements related to each sample is shown below it. **d,e,f**: Amplitudes of the $\pi$-periodic oscillations extracted from (d,e,f), for both longitudinal and transverse resistances, at various magnetic fields between 0.5 T and 1.5 T, and their field dependences are fitted to power laws (dashed lines).

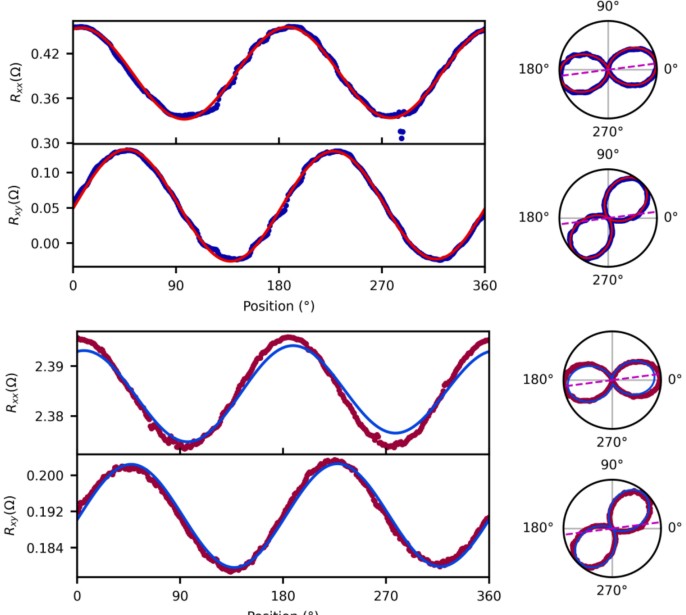

FIG. 4. **a,c**: Angular dependence at 14 T and 1.9 K (a) and 200 K (c) of the longitudinal ($R_{xx}$, top panels) and transverse ($R_{yx}$, bottom panels) resistances for sample D2, in Cartesian coordinates. Fits to the PHE model are shown in red (a) and blue (c). **b,d**: Same data as in **a,c**, in polar coordinates. The dashed-line represents the orientation of the current estimated from the data.

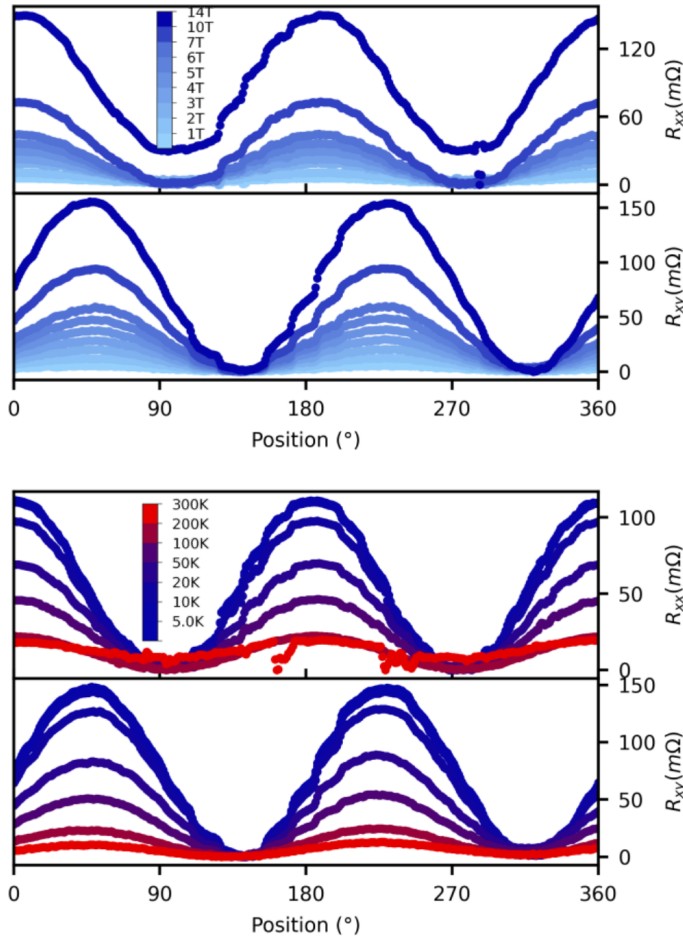

FIG. 5. **a**: Angular dependence of the longitudinal ($R_{xx}$, top panels) and transverse ($R_{yx}$, bottom panels) resistances of sample2 D2 at 5K and multiple fields from 1T to 14T. The plots are shifted vertically for visibility, to share a minimum at 0 Ω. **b**: Same as in **a**, for data taken at 14T and at multiple temperatures from 5K to 300K. The plots are shifted vertically for visibility, to share a minimum at 0 Ω.

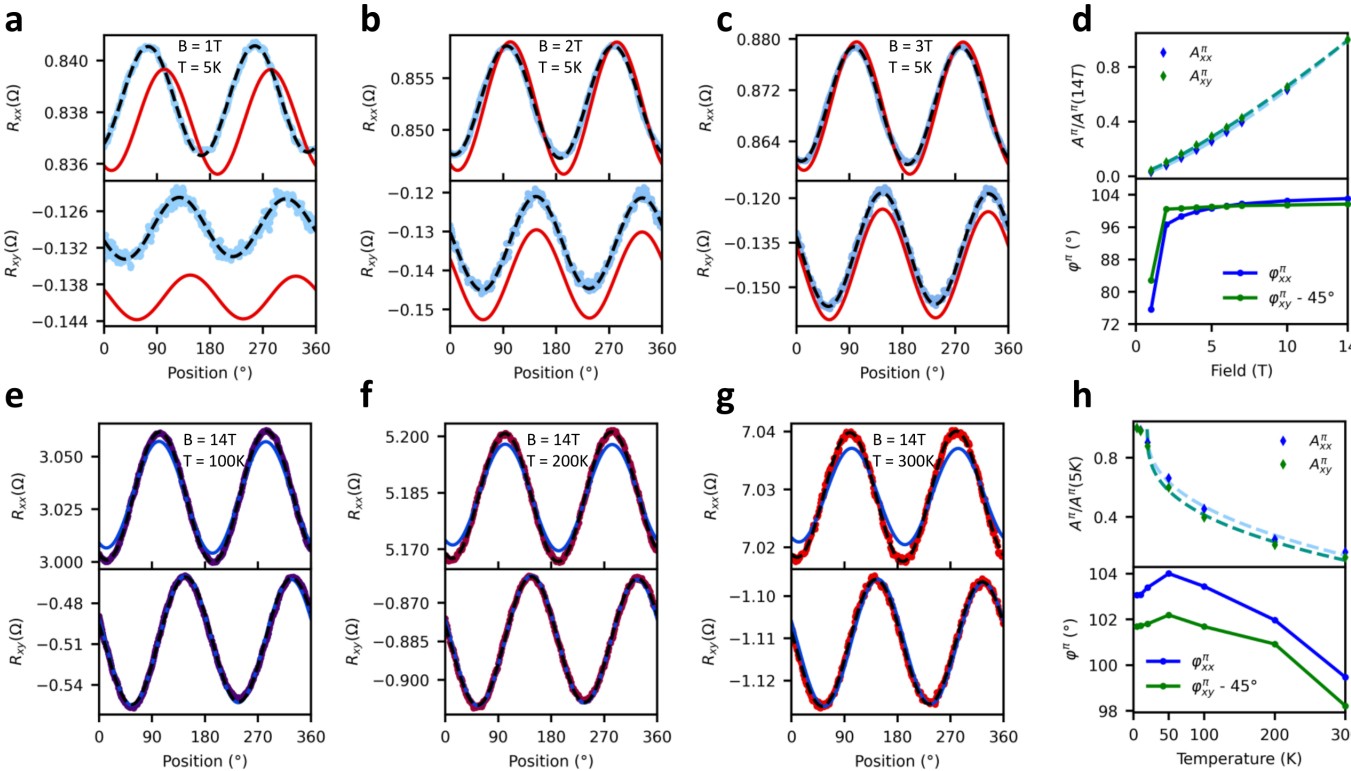

FIG. 6. **a,b,c**: Angular dependence of the longitudinal ($R_{xx}$, top panels) and transverse ($R_{yx}$, bottom panels) resistances of sample $D2$ at 1.9 K and 1T (a), 2T (b) and 3T (c). as the variation of the phase and the offset are significantly reduced compared to $D1$, the constrained fits of the PHE model (in red) doesn't deviate as much from the measurements at low fields. The data can be fitted very well with an unconstrained $\pi$-periodic fit (in dashed black). **d**: Field dependence of the amplitude (top panel, renormalized to its value at 14T) and phase (bottom panel) of the unconstrained fits of the longitudinal ($R_{xx}$, blue) and transverse ($R_{yx}$, green) resistances of $D2$. The renormalized amplitudes of the oscillations in $R_{xx}$ and $R_{yx}$ have the same field dependence, as expected for the PHE. The phase does not vary much in field, with an offset of about $45° + 6°$ between $\varphi_{xx}^{\phi}$ and $\varphi_{yx}^{\phi}$. **e,f,g**: Angular dependence of the longitudinal ($R_{xx}$, top panels) and transverse ($R_{yx}$, bottom panels) resistances at 14 T and 100 K (e), 200 K (f) and 300 K (g) for $D2$. The constrained fits of the PHE model (in blue) start deviating from the measurements in $R_{xx}$ at high temperature, as the amplitude of the oscillations are smaller than anticipated from those in $R_{yx}$, just as in $D1$. The data can still be fitted very well with an unconstrained $\pi$-periodic fit (in dashed black). **h**: Temperature dependence of the amplitude (top panel, renormalized to its value at 5 K) and phase (bottom panel) of the unconstrained fits of the longitudinal ($R_{xx}$, blue) and transverse ($R_{yx}$, green) resistances for $D2$. The renormalized amplitudes of the oscillations in $R_{xx}$ and $R_{yx}$ have a different dependence in temperature, as the amplitude in $R_{xx}$ decreases more slowly than that in $R_{yx}$. The phase of the oscillations changes slightly with the temperature.

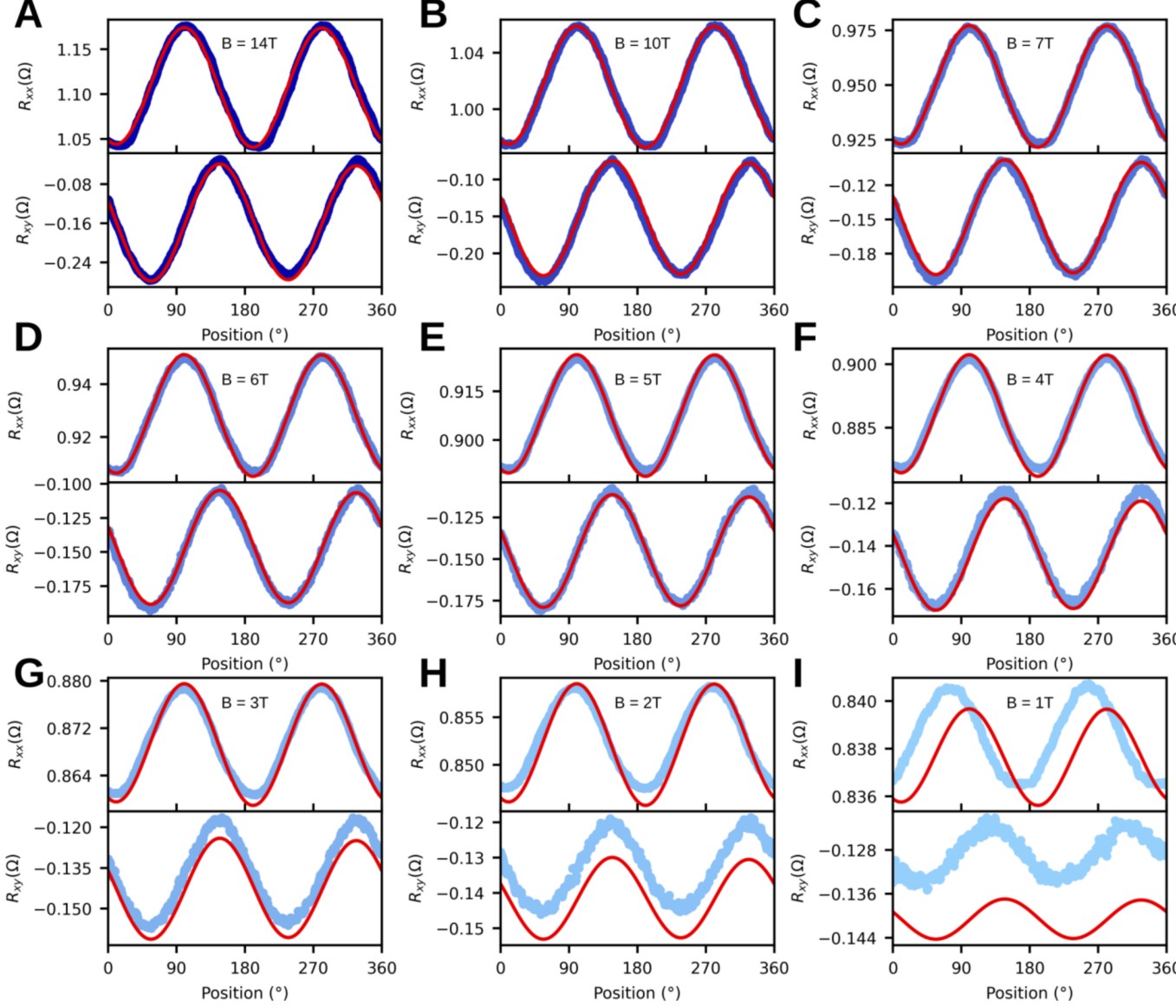

FIG. 7. Raw data: angular dependence of the resistance at different magnetic fields from 1T to 14T for sample D1, at T=5K. The red lines show the fits to the PHE model.

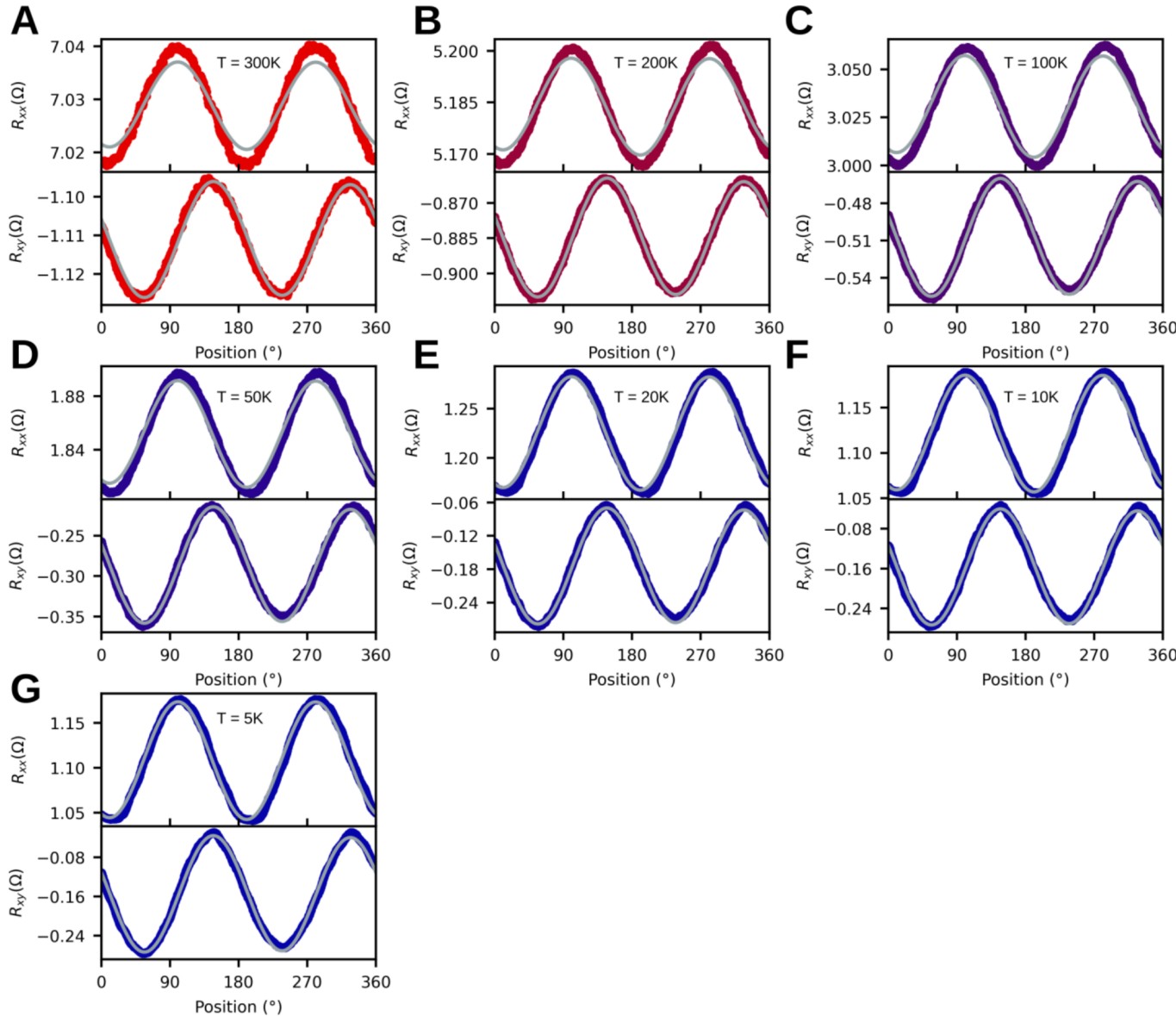

FIG. 8. Raw data: angular dependence of the resistance at different temperatures from 5K to 300K for sample D1, at B=14T. The grey lines show the fits to the PHE model.

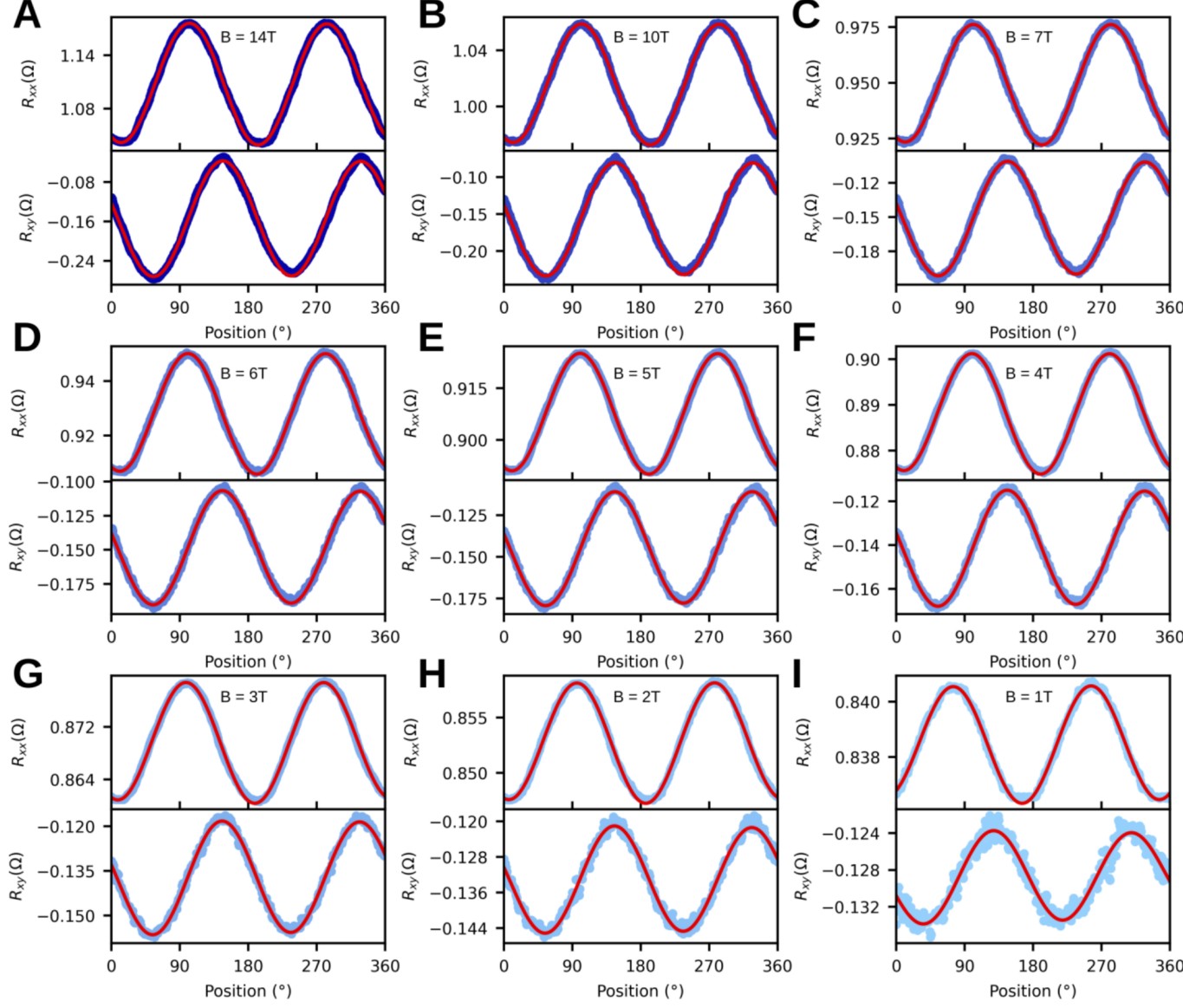

FIG. 9. Raw data: angular dependence of the resistance at different magnetic fields from 1T to 14T for sample D1, at T=5K. The red lines show the fits to the simple fit model.

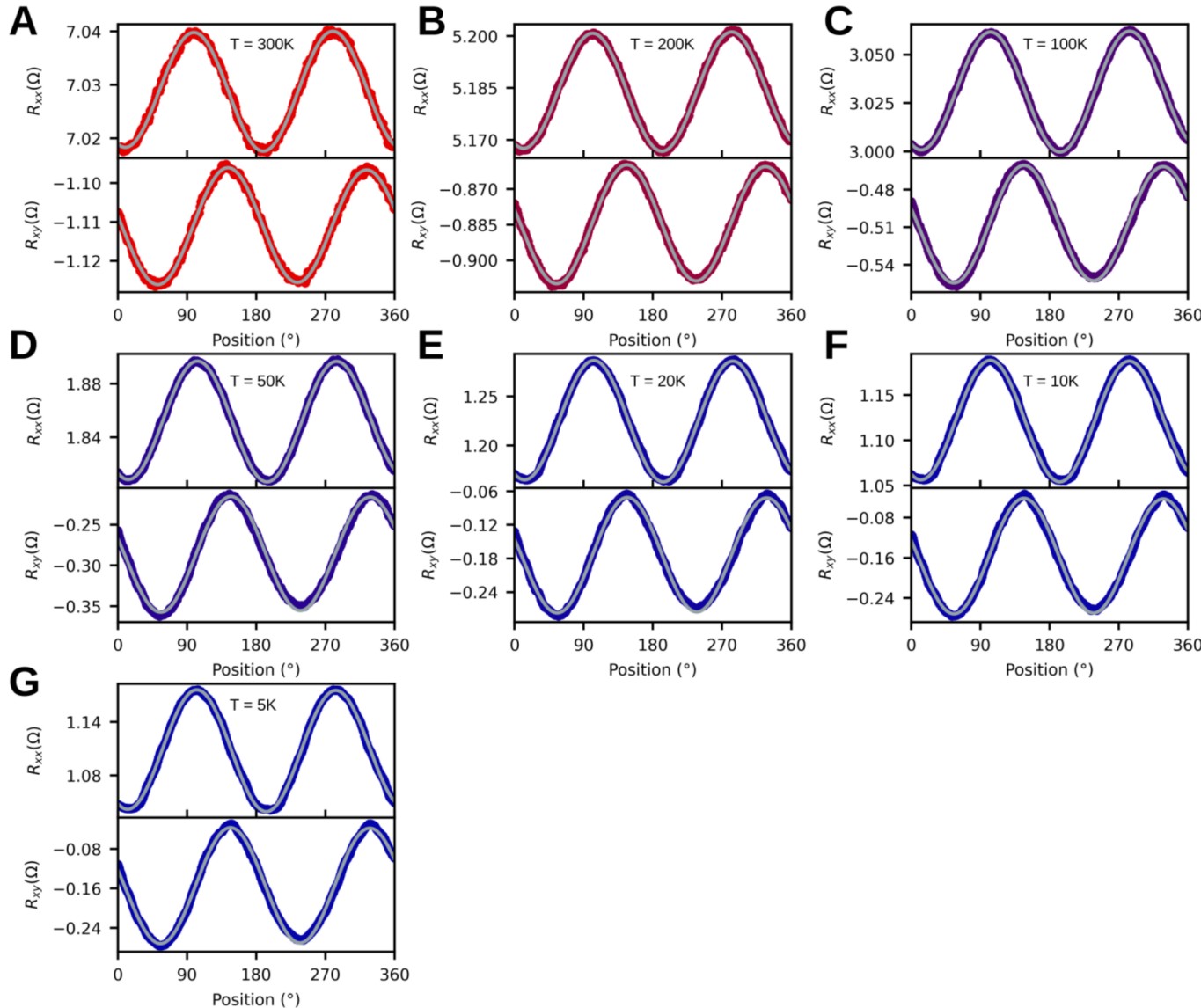

FIG. 10. Raw data: angular dependence of the resistance at different temperatures from 5K to 300K for sample D1, at B=14T. The grey lines show the fits to the simple fit model.