# Peer review of "Room temperature Planar Hall effect in nanostructures of trigonal-PtBi2"

_SciPost Physics_

## Round 2 · Referee Report · Anonymous (Referee 3) · 2025-4-16

Report

The authors have measured the transverse and the longitudinal resistance of three exfoliated samples of t-PtBi2 flakes in a magnetic field rotating the mutual orientation of applied current and magnetic field. The study produces interesting data and beautiful figures. However, this reviewer is not impressed by the analysis and the presentation and believes that it is not appropriate for Scipost. Here is the list of my objections.

i) The main observation is that both longitudinal and transverse resistance oscillate with a periodicity of 180° and these oscillations are phase shifted. But this is a trivial result, expected in any metal. When the magnetic field and the electric current, there is a Lorentz force and when they are parallel, there is no Lorentz force. Independent of microscopic details. When the current and field are parallel to each other, the longitudinal resistance is expected to be lower than when they are perpendicular. When the current and field are parallel to each other, the transverse resistance is expected to be zero, provided that: i) There is no misalignment; ii) The symmetry axes of the crystal and the Fermi surface pockets coincide with each other. There is no discussion of these prosaic possibilities in this paper.

ii) A serious study of transport in any metal (topological or otherwise) would inform the reader about the RRR of the samples, the mean free path (or the mobility) of electrons, the carrier density of the system, the shape and the anisotropy of the Fermi surface, before linking the results to any exotic physics, such as the presence of Weyl nodes. There is no trace of this here. The modest amplitude of the magnetoresistance indicates that low-temperature mobility is not very high.

iii) The issue of reproducibility is not addressed (or rather addressed in a strange way). Yes, all three samples show phase-shifted oscillations of the conductivity. But this is trivial. Do the color figures of Fig.3d-i, become similar when the same angular convention (phi=0 when I//B) is used? Why not plot resistivity (and not resistance)? Is the amplitude of the oscillations in three samples identical in resistivity or conductivity? This looks unlikely. The relative size of oscillations in Rxx is 4e-3 in sample 19e-3 in sample 2, and 22e-3 in sample 3. Since sample 2 and 3 are those which have the most irregular shape, one wonders if the observed signal is not enhanced by them. The irreproducibility in the case of the Hall resistance is even larger. Even the sign of the effect is not the same across the three samples.

iv) There is no information about the in-plane orientation of the applied current respective to the crystal axes. Is it the same in the three samples?

v) The main result appears in the title of the paper. Have the authors observed a planar Hall effect at room temperature? I am not convinced. One needs to exclude misalignment. Since the authors admit that the plane of rotation may not be the (electric field, charge current) plane, then the whole signal may be due to a combination of rotational misalignment and a genuine in-plane anisotropy of Fermi velocity. Only a two-axis rotational set-up can dissipate any ambiguity. The fact that the oscillations keep their phase from 1K to 300 K is actually an argument in favor of misalignment origin.

Recommendation

Reject

  • validity: -
  • significance: -
  • originality: -
  • clarity: -
  • formatting: -
  • grammar: -

Author:  Arthur Veyrat  on 2025-10-28  [id 5961]

(in reply to Report 1 on 2025-04-16)

We thank the reviewer for their input as to our manuscript.
You will find attached our point-by-point response, as well as a version of the paper with modifications highlighted (at the end of the response).

Attachment:

Veyrat_et_al_wQ7idfO._Scipost_Response_2__Main__SM_compressed.pdf

---

## Round 2 · Referee Report · Anonymous (Referee 1) · 2025-4-25

Report

I appreciate the authors' response to my technical queries, but I am not convinced by their response to some of the conceptual questions. To my understanding, the main claims of this manuscript include: i) the observation of the planar Hall effect up to room temperature; ii) experimental evidence of the impact of band topology on the electronic transport ("Our results strengthen the topological nature of PtBi2 and the strong influence of quantum geometric effects on the electronic transport...")

Regarding i), another reviewer has already raised some technical concerns that have to be addressed. My main doubts concern the point ii), as follows:

  1. Fig. 1 depicting 12 Weyl nodes has been added to the manuscript. However, other Weyl nodes are still prominently mentioned in the discussion. Some of them lie very far away from the Fermi level (-655 meV, -497 meV), and it is very hard to imagine that such features may contribute to the transport, so I am not sure why they are mentioned at all. It gets even more confusing with the last added sentence in the first paragraph of the Discussion: "...these 6 groups are referred to as the field-generated Weyl nodes, as they exist even in the absence of an external magnetic field". It's probably a typo, and the authors imply that these nodes appear in the magnetic field only. But on reading this paragraph several times I am still not quite sure what the intended meaning is. The whole discussion of these "distant" Weyl nodes is very obscure and seemingly irrelevant to the experimental data presented in this work, especially if APHE is excluded.

  2. Different calculations of the Fermi surface (Fig. 8c in PRM'2020; Fig. 1c in Nature'2024; Fig. 5 in arXiv:2504.13661) all show large sheets that lie somewhat far away from the anticipated Weyl nodes shown in Fig. 1 of the present work. What is the rationale for discussing electronic transport in the context of these Weyl nodes? What about the large and "conventional" parts of the Fermi surface that should, naively, dominate the transport?

  3. A concurrent study of the planar Hall effect in PtBi2 has been published by Zhu et al. [PRB 110, 125148 (2024)]. It appeared in September 2024, about a month before the initial submission and half a year before the resubmission of the present manuscript, yet it has not been mentioned at all. Detailed transport measurements are certainly time-consuming, and the publication of similar data by another group does not compromise the novelty per se. However, it does raise some conceptual questions about the interpretation, because Zhu et al. arrive at a very different conclusion that the properties of the PtBi2 flakes are dominated by anisotropic orbital magnetoresistance. In fact, that study looks more convincing because it juxtaposes transport measurements on the bulk samples and thin flakes, and eventually identifies the possible effects of band topology in the bulk but not in the flakes. It also shows a direct comparison of the RRR of the bulk and thin-flake samples and demonstrates a quite drastic reduction in the sample quality (in terms of RRR) upon exfoliation. While a similar reduction in RRR can be inferred from the authors' response, it does not appear prominently in the manuscript, although it seems crucial for the interpretation, especially with the knowledge that bulk samples show a much stronger resemblance to the expected behavior of a Weyl semi-metal than the thin flakes.

As a peer, I find it confusing and even disconcerting that similar data are used to produce entirely different claims: orbital magnetoresistance by Zhu et al. vs. "quantum geometric effects" in the present work. In my opinion, the authors should either find an interpretation consistent with Zhu et al., or provide clear-cut arguments why their data unambiguously prove the role of band topology in the electronic transport. Otherwise, this work will mainly generate confusion instead of clarifying the physics of the potentially interesting quantum material.

Additionally, I would like to mention that I find it rather disturbing when different pieces of characterization are scattered across different publications. Readers are sent to Ref. 19 to see temperature dependence of the resistivity and RRR, and to the supplemental material of the Nano Lett.'2023 publication to see the effect of the contacts geometry. It is at best inconvenient. Moreover, some of the most exciting results for PtBi2 demonstrate an acute sample dependence, see the very unsystematic superconducting gaps shown in the STM study [Nature Comm. 15, 9895 (2024)] and the absence of superconductivity in arXiv:2503.08841, as opposed to Nature'2024, etc. While I understand that these different Dresden-centered publications on PtBi2 are related to different collaborators, I have to say that there are growing doubts on whether all unconventional and exciting physics of PtBi2 is really intrinsic. Careful tracking of the sample characterization and comparisons between the bulk and thin-flake samples could help to resolve these doubts.

Recommendation

Ask for major revision

  • validity: -
  • significance: -
  • originality: -
  • clarity: -
  • formatting: -
  • grammar: -

Author:  Arthur Veyrat  on 2025-10-28  [id 5960]

(in reply to Report 2 on 2025-04-25)
Category:
answer to question

We want to thank the reviewer for their once again very thorough review.
You will find attached our point-by-point response, as well as a version of the paper with modifications highlighted (at the end of the response).

Attachment:

Veyrat_et_al._Scipost_Response_2__Main__SM_compressed.pdf

---

## Round 2 · Author Response

Following the very constructive comments by the esteemed reviewers, we have now improved our manuscript (see point-by-point response).
We note that an author (Iryna Kovalchuk, who contributed to the crystal growth) was mistakenly omitted in the previously submitted version. This mistake has been fixed in the present version.

---

## Round 2 · List of Changes

We have implemented multiple modifications to the manuscript in response, mainly by clarifying the link between this study and Ref. 19 (now Ref. 12), more details on the analysis, more clarity on the different Weyl nodes in the system, a dedicated Methods section, and additional details about the samples.
We also added more information about the samples (in methods and SM), we show the measurements from additional contact configurations (in SM), provide a more detailed discussion of the Weyl nodes contribution to the PHE, and clarify the link between this study and Reference 19 (now Ref. 12, in the introduction).
For more details, see the response to reviewers.

---

## Round 3 · Referee Report · Anonymous (Referee 3) · 2025-11-3

Report

In their response the authors write: “The resistivity is not the same between the three samples, nor is the amplitude of the oscillations in resistivity the same. We have no clear explanation for that but we note however that a detailed analysis can be rather complicated by the fact that the RRR is strongly thickness dependent (unexplained so far but also observed by other groups) and by the contribution of surface states.”

“The fact that RRR is strongly thickness dependent” is widespread among metallic solids. It not only implies that the electronic mean free path in nanoflakes is much shorter than in bulk samples but that the path of charge carriers is less controlled. This opens the way for what the investigators of bulk transport in layered conductors call “c-axis contamination”.

Figure 17 of the present version reveals that even the temperature dependence of zero-field resistance in these low RRR samples is not reproducible.
How can one base any solid conclusion by studying these samples?
R _xx and R_xy oscillations (seen in Fig.2j,k,l) have similar amplitudes. This can happen if an electric current with uncontrolled orientation generates tan electric field of similar amplitude in the two channels. with a small shift between them.

This interpretation is backed by the comparison shown in Figure 5 of PRB 110, 125148 (2024). The bulk sample shows a significant magnetoresistance with visible structure as expected in a metal with a non-spherical Fermi surface and reasonably mobile carriers. In contrast, the nanoflakes show small featureless magnetoresistance and their Hall resistivity mirrors the weak magnetoresistance.

If the "room temperature planar Hall effect" is a genuine feature why should it disappear in cleaner samples?

Occam’s razor invites us to look for the simplest available explanation of any observation.

Recommendation

Reject

---

## Round 3 · Referee Report · Anonymous (Referee 1) · 2025-11-22

Report

I would like to thank the authors for addressing my previous queries. I appreciate the differences between their study and the previous work from PRB 110, 125148 (2024). I also acknowledge that more extensive transport measurements are reported in the present case compared to earlier studies. However, I still do not understand which features make the authors believe that their data "strengthen the topological nature of PtBi2". I found the corresponding part of the reply rather obscure, and I can not support the chosen wording "the measurements are entirely consistent with the prediction of Weyl topology", because it obviously works in the other direction too. The measurements are entirely consistent with the anisotropic orbital magnetoresistance scenario as well.

Two additional reasons for my doubts are some recent publications on t-PtBi2:

  1. [Appl. Phys. Lett. 126, 233101 (2025)] from June 2025 argues for the Fermi surface anisotropy as the origin of the Hall response

  2. [Phys. Rev. Materials 9, 084202 (2025)] reports Hall and Nernst effect on single crystals. This paper contains explicit statements against the role of topology in t-PtBi2. For example: "Weyl nodes are a natural source of Berry curvature... and explain ANE contribution even in nonmagnetic compounds such as Cd3As2 and TaP. Although a similar mechanism cannot be completely excluded a priori in PtBi2, our material presents substantial differences with respect to the mentioned cases". The behavior of PtBi2 is then attributed to a "multiband picture" and not to the Weyl nodes.

The authors of that paper further argue that "Weyl nodes in t-PtBi2 are predicted to be located at about 47 meV above the Fermi level, which may be too far to make them dominate the Nernst effect with an anomalous component". Why would the very same Weyl nodes determine the Hall effect but play no role in the Nernst effect of the material?

The aforementioned paper concludes with the statement "...our study does not reveal any evident contribution related to nontrivial topology". Ironically, some of the present authors are co-authors of that paper too.

While the present manuscript on the planar Hall effect in t-PtBi2 contains potentially interesting data, I can not support its publication because it would only lead to a major confusion. Unless I am missing something, the authors seem to have chosen an opportunistic approach and make inconsistent statements regarding the role of topology in the material. I do not think that community would benefit from such a series of mutually contradicting scenarios of t-PtBi2.

Recommendation

Ask for major revision

---

## Round 3 · Author Response

Following the very constructive comments by the esteemed reviewers, we have now improved our manuscript (see list of changes).
We have also provided a point-by-point response to the comments of the reviewers on the previously submitted version.

---

## Round 3 · List of Changes

• The discussion on the possible relative importance of the different topological features to the PHE was shifted to the supplementary materials (SM), to focus on the main point of the manuscript and avoid possible confusion.
  • At the request of a reviewer, additional information was added to the SM, adapted (with permission) from our previous publications, to centralize information as much as feasible.
  • Details were added to the description of the PHE to make explicit that, despite its name, the PHE is not associated with a Lorentz force. -Minor typos were corrected.

---

## Editorial Decision

awaiting_resubmission